# SHE-LoRA: Selective Homomorphic Encryption for Federated Tuning with Heterogeneous LoRA

**Jianmin Liu**[1], **Li Yan**[1]*, **Borui Li**[1], **Lei Yu**[2], **Chao Shen**[1]
[1]Xi'an Jiaotong University  [2]Rensselaer Polytechnic Institute
jianmin.liu@stu.xjtu.edu.cn, li.yan.88@xjtu.edu.cn,
boruili@stu.xjtu.edu.cn, yul9@rpi.edu, chaoshen@mail.xjtu.edu.cn

## Abstract

Federated fine-tuning is critical for improving the performance of large language models (LLMs) in handling domain-specific tasks while keeping training data decentralized and private. However, prior work has shown that clients' private data can actually be recovered via gradient inversion attacks. Existing privacy preservation techniques against such attacks typically entail performance degradation and high costs, making them ill-suited for clients with heterogeneous data distributions and device capabilities. In this paper, we propose SHE-LoRA, which integrates selective homomorphic encryption (SHE) and low-rank adaptation (LoRA) to enable efficient and privacy-preserving federated tuning of LLMs in cross-device environments. Based on model parameter sensitivity assessment, heterogeneous clients adaptively negotiate and select a subset of model parameters for homomorphic encryption. To ensure accurate model aggregation, we design a column-aware secure aggregation method and customized reparameterization techniques to align the aggregation results with the heterogeneous device capabilities of clients. Extensive experiments demonstrate that SHE-LoRA maintains performance comparable to non-private baselines, achieves strong resistance to state-of-the-art attacks, and significantly reduces communication overhead by 99.71% and encryption time by 99.87%, compared to HE baselines.

## 1 Introduction

Large language models (LLMs) have excelled in various tasks, but their deployment in domain-specific applications (e.g., healthcare, finance) often requires private, user-generated data (Durante et al., 2024; Huang et al., 2024). However, stringent privacy preservation regulations like GDPR (Voigt & Von dem Bussche, 2017) pose significant barriers to centralized fine-tuning on such data. To address this, federated learning (FL) emerged as a promising solution by enabling decentralized parameter-efficient fine-tuning (PEFT) of LLMs without exposing raw data (Zhang et al., 2024).

Among various PEFT techniques, Low-Rank Adaptation (LoRA) stands out due to its high efficiency and model quality. It reparameterizes the weight matrix $\mathbf{W} \in \mathbb{R}^{m \times n}$ as $\mathbf{W} = \mathbf{W_0} + \Delta\mathbf{W} = \mathbf{W_0} + \mathbf{BA}$, where $\mathbf{W_0} \in \mathbb{R}^{m \times n}$ represents the frozen pre-trained parameters, and $\mathbf{B} \in \mathbb{R}^{m \times r}$ and $\mathbf{A} \in \mathbb{R}^{r \times n}$ are the two low-rank adapter matrices to be learned. Given that the rank $r \ll \min(m, n)$, LoRA significantly reduces both computation and communication costs in federated PEFT. However, recent works (Petrov et al., 2024; Balunovic et al., 2022) have shown that the parameters or gradients transmitted during federated PEFT can be exploited via *inversion attacks* to reconstruct private training data, highlighting the need for stronger privacy protection in federated PEFT with LoRA.

Many privacy-preserving techniques have been proposed to mitigate privacy leakage risks in FL, including differential privacy (DP) (Sun et al., 2024c; Yu et al., 2022; Zhu et al., 2025), secure multi-party computation (MPC) (Mugunthan et al., 2019; Kanagavelu et al., 2020; Zheng et al., 2024)), and homomorphic encryption (HE) (Han & Yan, 2023; Jin et al., 2023; Hu & Li, 2024)). DP ensures formal privacy guarantees by perturbing data or model updates with random noise. However, in LoRA-based settings that involve the multiplication of $\mathbf{A}$ and $\mathbf{B}$, this noise becomes amplified through the multiplication, often hindering convergence and degrading model performance (Sun et al., 2024c). In contrast, cryptographic approaches such as MPC and HE can achieve higher

---

*Li Yan is the corresponding author of this paper.
Code is publicly available at https://github.com/liyan2015/SHE-LoRA.

accuracy. MPC-based secure aggregation employs techniques like garbled circuits and secret sharing to securely compute PEFT updates. Nevertheless, it often requires intricately designed computation and synchronization protocols, making it less practical for FL with heterogeneous data and device capabilities (Kairouz et al., 2021; Li et al., 2020). Selective HE (SHE) (Han & Yan, 2023; Jin et al., 2023; Hu & Li, 2024) offers a compelling alternative by encrypting only sensitive parameters and allowing computation over ciphertexts, delivering strong privacy guarantees with low HE costs and preserving accuracy for privacy-preserving federated PEFT.

However, existing SHE methods struggle to balance privacy and efficiency in cross-device federated PEFT with LoRA, particularly under Non-IID (Non-Independently Identically Distributed) data and heterogeneous device capabilities. As discussed in Section 2.4, two observations highlight these challenges: 1) LoRA matrix multiplication makes $\Delta \mathbf{W}$ denser, which may increase the number of parameters requiring encryption, and 2) heterogeneous clients produce different encrypted parameter positions, whose union during aggregation expands the encrypted set and inflates HE costs. Driven by these limitations, we aim to adaptively balance security and HE overhead per client in cross-device federated PEFT with LoRA. Achieving this goal requires addressing the following key challenges:

• **How to adaptively apply SHE across heterogeneous clients?** Algorithms like FedAvg (McMahan et al., 2017) are not directly applicable to LoRA-based PEFT. Naively applying LoRA requires all clients to use the same low-rank configuration, which is impractical for devices with heterogeneous capabilities. Furthermore, separately aggregating the adapter matrices ($\mathbf{A}$ and $\mathbf{B}$) is not mathematically equivalent to full-weight aggregation (Section 2.4). Heterogeneous hardware also prevents clients from encrypting updates of the same size, since clients with limited resources can encrypt only a small subset of parameters while stronger devices may encrypt more. This mismatch disrupts aggregation and complicates reconstruction of low-rank matrices from encrypted weights. Thus, a new aggregation algorithm that is efficient, accurate, and compatible with SHE is needed.
• **How to avoid expansion of encrypted subsets under SHE?** In heterogeneous settings, clients may independently encrypt arbitrary positions in their model parameter matrix, inflating ciphertext size during aggregation. Moreover, mixing plaintext and ciphertext matrices introduces structural disorder, making aggregation inefficient. Without coordinated negotiation of encryption positions, both ciphertext size and aggregation overhead can increase substantially.

To address these challenges, we propose SHE-LoRA, which integrates SHE and LoRA to enable efficient and privacy-preserving federated PEFT in cross-device environments. Specifically,

• We devise a HE subset negotiation mechanism that tailors model-parameter encryption to each client's capabilities. Each client assesses its model parameter importance and selects an affordable subset for HE based on its resource constraints and privacy needs. This subset is encoded using order-preserving encryption (OPE) and sent to a server, which then negotiates a global HE subset to optimally balance privacy and HE overhead across heterogeneous clients.
• We introduce a selective parameter encryption method based on column-swapping parameter obfuscation, which clusters unencrypted and encrypted parameters separately, enabling efficient matrix operations on plaintexts and batch encryption of ciphertexts. Moreover, obfuscating encrypted parameter positions increases adversarial uncertainty and mitigates privacy leakage.
• We propose a column-aware adaptive aggregation method, which aligns encrypted columns across clients for efficient and accurate aggregation of adapter matrices and subsequent reparameterization to recover LoRA parameters without losing meaningful model updates.
• Experiments on clients and LLMs with varying scales demonstrate that SHE-LoRA provides strong resistance to state-of-the-art (SOTA) attacks while maintaining model performance comparable to non-private baselines. Compared to HE baselines, SHE-LoRA reduces communication overhead by up to 99.71% and HE overhead by 99.87%.

## 2 PRELIMINARIES AND MOTIVATIONS

### 2.1 DEFINITION OF PARAMETER SENSITIVITY

Inspired by model pruning, which removes unimportant model parameters to reduce model size while maintaining performance, SHE identifies and selectively encrypts the most important parameters. Specifically, given model parameters $\mathbf{W}$, let $\mathcal{L}(\mathbf{W})$ denote the loss function. For a subset of the model parameters $\mathbf{w} \in \mathbf{W}$, and the model parameters with $\mathbf{w}$ zeroed out (denoted as $\mathbf{W}_{-\mathbf{w}}$), the sensitivity of $\mathbf{w}$ is defined as the change in loss when $\mathbf{w}$ is zeroed out:

$$\Omega(\mathbf{w}) = |\mathcal{L}(\mathbf{W}) - \mathcal{L}(\mathbf{W}_{-\mathbf{w}})|. \tag{1}$$

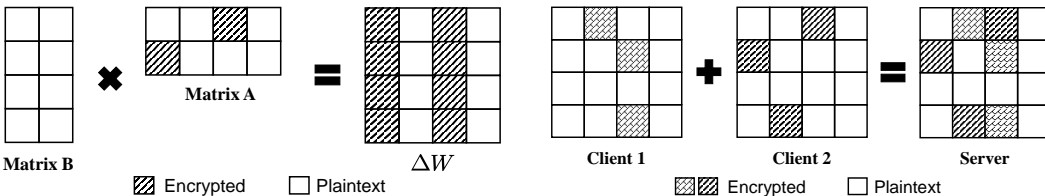

Figure 1: Expansion of encryption positions.    Figure 2: Inflation of ciphertext size.

Eq. (1) implies that a larger loss change upon removing $\mathbf{w}$ indicates higher sensitivity of $\mathbf{w}$. Thus, $\Omega(\mathbf{w})$ reflects not only importance, but also the potential privacy leakage risk associated with exposing $\mathbf{w}$. However, directly computing $\Omega(\mathbf{w})$ for all parameters is computationally infeasible. Taylor approximation-based estimation (Frantar et al., 2023) requires gradient computation, which also incurs significant overhead, especially in LLMs, where high dimensionality and outlier activations further exacerbate the cost (Sun et al., 2024a). To mitigate this, we adopt Wanda (Sun et al., 2024b) to estimate the sensitivity of a parameter $W_{ij}$ in the $i$-th row and $j$-th column as:

$$\Omega(W_{ij}) = |W_{ij}| \cdot \|x_j\|_2, \tag{2}$$

where $|\cdot|$ is the absolute value operator, $\|x_j\|_2$ is the $l_2$ norm of the $j$-th features in the input data $\mathbf{X}$. Since both $\mathbf{W}$ and $\mathbf{X}$ are directly accessible, Wanda estimates parameter sensitivity with only a single forward pass. See Appendix B.1 for details on the link between high-sensitivity parameters and privacy risk.

## 2.2 PRIVACY LEAKAGE QUANTIFICATION

We leverage mutual information to quantify privacy leakage caused by the selective encryption of $\mathbf{w}$. Specifically, we assume that once encrypted with HE, $\mathbf{w}$ does not leak any privacy information. While for the accessible plaintext portion of the model parameters (i.e., $\mathbf{W}_{-\mathbf{w}}$), we measure the mutual information shared between $\mathbf{W}_{-\mathbf{w}}$ and $\mathbf{W}$ as:

$$I(\mathbf{W}; \mathbf{W}_{-\mathbf{w}}) = \sum_{y \in \mathbf{W}_{-\mathbf{w}}} \sum_{x \in \mathbf{W}} p(x, y) \log_2 \frac{p(x, y)}{p(x)p(y)}, \tag{3}$$

where $p(x, y)$ is the joint probability distribution, $p(x)$ and $p(y)$ are the marginal distributions of $\mathbf{W}$ and $\mathbf{W}_{-\mathbf{w}}$, respectively. $I(\mathbf{W}; \mathbf{W}_{-\mathbf{w}})$ quantifies the extent of privacy leakage attributable to the unencrypted model parameters. As the value of $I(\mathbf{W}; \mathbf{W}_{-\mathbf{w}})$ increases, the risk of privacy leakage due to the selective encryption of $\mathbf{w}$ also increases. See Appendix F.3 for implementation details.

## 2.3 THREAT MODEL

Following prior works (Jin et al., 2023; Kiani et al., 2025), we consider a semi-honest adversary $\mathcal{A}$ that may compromise the aggregation server or a subset of clients. While $\mathcal{A}$ follows the training protocol, it passively attempts to infer client data from observed model updates. We assume that 1) when $\mathcal{A}$ compromises a subset of clients, it can only infer private information from the clients' local models; 2) when $\mathcal{A}$ compromises the aggregation server, it can only infer private information from unencrypted parameters; 3) when both the aggregation server and a subset of clients are compromised, $\mathcal{A}$ can access the private key (shared among all clients) to decrypt model updates sent from benign clients, which can be addressed via multi-party HE techniques such as multi-key HE, proxy re-encryption, etc. The multi-party HE techniques and protection against other malicious behaviors (e.g., poisoning, backdoor attacks) are not the focus of this work, and we refer to existing defenses (Zheng et al., 2022; Queyrut et al., 2023) and possible extensions in Appendix C as future endeavors.

## 2.4 MOTIVATIONS

**Naive averaging of LoRAs leads to mathematical errors.** Popular federated LoRA methods (Zhang et al., 2024; Yan et al., 2024; Meng et al., 2024; Babakniya et al., 2023) require clients to possess homogeneous LoRA ranks, and aggregate $\mathbf{A}$ and $\mathbf{B}$ separately across clients (i.e., server side $= \sum \mathbf{B} \times \sum \mathbf{A}$, $r_A = r_B$). However, this introduces inconsistency in global model updates, as the aggregation of LoRA updates (i.e., $\sum(\mathbf{B} \times \mathbf{A})$) is intrinsically unequal to $\sum \mathbf{B} \times \sum \mathbf{A}$, which will degrade model performance. Moreover, separately aggregating the LoRA matrices requires all clients to use the same rank, which is unrealistic for cross-device federated LoRA. Thus, these naive approaches are inapplicable to heterogeneous LoRA settings.

**Matrix multiplication expands encryption positions.** Albeit the strong privacy guarantee of HE, applying HE per parameter is computationally and communicatively expensive. Although existing SHE methods like FedML-HE (Jin et al., 2023) and MaskCrypt (Hu & Li, 2024) reduce HE overhead

by selectively encrypting a subset of model parameters with masking, the matrix multiplication of LoRA will lead to an expanded HE subset as shown in Fig. 1, which significantly impairs the cost-saving performance of SHE for federated LoRA.

**Matrices from heterogeneous clients inflate ciphertext size.** Our experiments show that clients with heterogeneous hardwares and Non-IID data tend to focus on different sensitive model parameters during fine-tuning. As a result, the positions selected for encryption vary across clients. In the aggregation phase of existing SHE methods, if a client encrypts a specific model parameter, the parameter corresponding to the same position must remain encrypted in the global model as well, leading to inflated ciphertext size as the number of clients grows, which is illustrated in Fig. 2.

## 3 METHOD

In order to adaptively balance security and HE overhead per client in cross-device federated LoRA, encrypting **A** offers a cost-effective solution. Since **A** directly operates on user data, it is more vulnerable to inversion attacks (Petrov et al., 2024), making its protection essential for preventing privacy leakage. Following this rationale, the diagram of SHE-LoRA is as illustrated in Fig. 3.

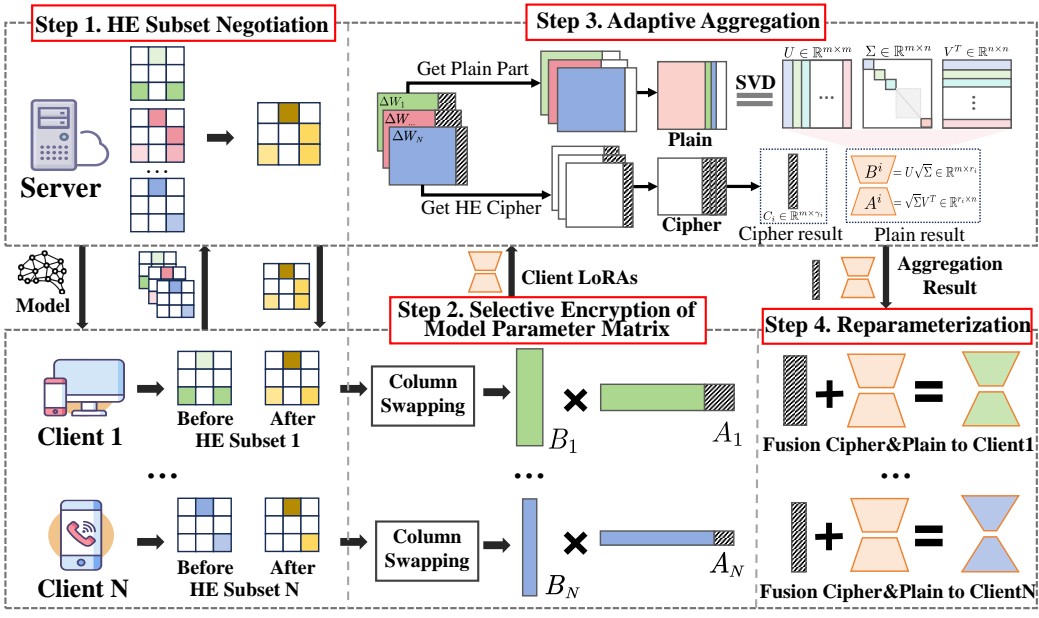

Figure 3: The workflow of SHE-LoRA.

SHE-LoRA consists of the following components: **Step 1. HE Subset Negotiation**. Based on the definition of model parameter sensitivity in Eq. (2), each client assesses and transmits its model parameter importance to a server. Then, the server negotiates a global HE subset and feeds it back to clients. **Step 2. Selective Encryption of Model Parameter Matrix**. Based on the global subset, clients perform column swapping to separately cluster unencrypted and encrypted parameters, enabling efficient matrix operations on plaintexts, batch encryption of ciphertexts, and parameter position obfuscation that enhances privacy protection and HE efficiency. **Step 3. Adaptive Aggregation**. The server performs adaptive, column-aware aggregation of the clients' unencrypted and encrypted parameters, respectively, enabling efficient and accurate aggregation of adapter matrices. **Step 4. Reparameterization**. Each client reparameterizes the aggregated plaintext and ciphertext results into LoRA parameters, matching its local rank for the next round of model tuning.

### 3.1 HE SUBSET NEGOTIATION

#### 3.1.1 ASSESSMENT OF MODEL PARAMETER IMPORTANCE

Fig. 4 shows the sensitivity of parameters measured by Eq. (2), where darker color indicates higher importance. We find that the sensitivity values generally differ by columns, suggesting a strong correlation with specific data channels. Considering that encrypting even a single element in a column will result in the expansion of encryption positions for that entire column due to matrix multiplication (Section 2.4), and vectorized encryption by columns is bene-

ficial to improving HE efficiency (Cheon et al., 2017), we let each client assess parameter importance by columns and determine its HE subset of columns based on its encryption budget.

Specifically, the encryption budget of Client $i$ is defined as the ratio of parameters for SHE (denoted as $\gamma_i \in [0, 1]$), which is specified according to its hardware capabilities such as CPU clock speed, etc. For an adapter matrix $\mathbf{A} \in \mathbb{R}^{r \times n}$ and input $\mathbf{X} \in \mathbb{R}^{L \times n}$, we use $S_j = \sum_{k=0}^{r} |\mathbf{W}_{kj}| \cdot \|x_j\|_2$ to calculate the importance of the $j$-th channel, where $\|x_j\|_2$ is the $l_2$ norm of the $j$-th feature $x_j \in \mathbb{R}^L$. Thus, the proposed approach can not only provide channel-level privacy protection, but also prevent unnecessary expansion of encryption positions.

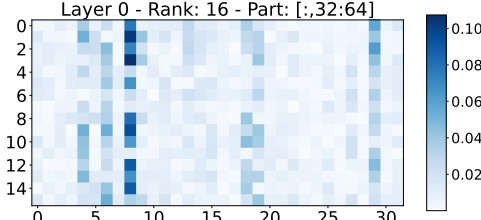

Figure 4: Sensitivity of model parameters.

### 3.1.2 HE SUBSET NEGOTIATION

As explained in Section 2.4, clients with heterogeneous data distributions and device capabilities will select different positions and amounts of important model parameters for encryption according to their individual budgets. As the union of HE subsets expands with the aggregation of client model updates, the ciphertext size may inflate significantly. To address this, we devise a HE subset negotiation mechanism to tailor client-specific encryption of model parameters, ensuring that the overall ciphertext size remains affordable per client.

To prevent the server from snooping on the positions of important model parameters and conducting targeted attacks, we apply OPE to hide each client's HE subset. Since SHE-LoRA is modular, OPE can be replaced by alternatives like order-revealing encryption, secure multi-party computation, or a trusted third party whenever stronger order privacy is required. As OPE only preserves numerical ordering of the plaintext, the server cannot obtain any information from the cipher other than the plaintext order. Specifically, Client $i$ first encrypts a tuple $(G_i, S_i)$ with OPE and sends it to the server, where $G_i$ is the set of columns that needs HE, and $S_i$ is their sensitivities. Then, the server maintains two shared lists based on all the clients' tuples: the *Common* list, which sorts all columns in $\bigcup_i G_i$ from most to least frequently deemed as sensitive, and the *Sensitivity* list, which sorts all columns in $\bigcup_i G_i$ from highest to lowest sensitivity. Finally, by taking into account both the overlap of important HE subsets across clients (reflected in *Common* and *Sensitivity*) and the preferred HE subsets of individual clients (reflected in $G_i$), the server negotiates a global HE subset affordable for each client, of which algorithmic details are elaborated in Appendix D.1. As a result, the negotiated global HE subset optimally balances privacy and HE overhead per client.

### 3.2 SELECTIVE ENCRYPTION OF MODEL PARAMETER MATRIX

As illustrated in the top of Fig. 5, the selected columns for encryption may be scattered across the parameter matrix. This irregular distribution increases the complexity of matrix batching and the overhead of encryption, decryption and computation. To address this, based on the negotiated HE subset, we propose a column-swapping method to separately cluster the columns pending for encryption and those remain unencrypted. This approach brings three key benefits: (1) encrypted columns are clustered together, allowing for efficient batch encryption with reduced storage and communication overhead; (2) the clustered unencrypted columns can be directly used in matrix operations, improving computational efficiency; and (3) the column-wise obfuscation increases the difficulty of potential privacy attacks.

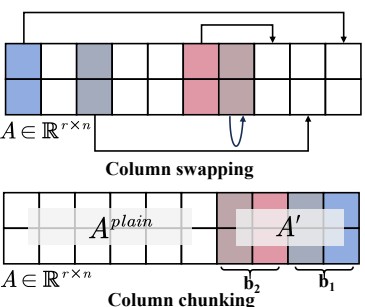

Figure 5: Selective encryption of columns.

After swapping, Client $i$ selectively encrypts its parameter matrix, and uses the last $k_i = \lfloor n \times \gamma_i \rfloor$ columns as its HE subset, which is affordable according to the client's encryption budget. Thus, the tensor to be encrypted, denoted as $\mathbf{A}'$, has the shape of $(r, k_i)$. In HE, the CKKS (Cheon-Kim-Kim-Song) scheme (Cheon et al., 2017) serves as an encryption technique that facilitates approximate floating-point arithmetic, making it particularly advantageous for the encryption of matrices or tensors. For large-scale tensors, the limited capacity of HE requires them to be partitioned into blocks, each encrypted separately. As illustrated in the bottom of Fig. 5, given a block size of *chunk*, $\mathbf{A}'$ can be

divided into $N^b = \lceil k_i/chunk \rceil$ blocks by column (denoted as $\mathbf{A}' = \{b_1, \ldots, b_j, \ldots, b_{N^b}\}$), where each block contains a tensor that has the shape of $(r, chunk)$. For each block $b_j$, the client applies CKKS encryption to obtain its ciphertext $C_j = \text{CKKS}(b_j, pk)$, where $pk$ is the public HE key. After all blocks are encrypted, the complete list of ciphertext blocks $\{C_j\}^{N^b}$ is sent to the server.

### 3.3 ADAPTIVE AGGREGATION

To prevent the inflation of ciphertext size caused by aggregating heterogeneous HE subsets as illustrated in Section 2.4, we propose an adaptive column-aware aggregation method to aggregate the unencrypted and encrypted parts separately.

**Aggregation of Unencrypted Model Parameters:** Upon receiving the adapter matrix $\mathbf{B}_i$ and the unencrypted part of $\mathbf{A}_i$ (denoted as $\mathbf{A}_i^{\text{plain}}$) from Client $i$, the server calculates the unencrypted weight update from Client $i$ as $\Delta\mathbf{W}_i^{\text{plain}} = \mathbf{B}_i\mathbf{A}_i^{\text{plain}}$. However, as the number of encrypted columns (i.e., $k_i$) differs across the clients due to their diverse encryption budgets, the shape of $\Delta\mathbf{W}_i^{\text{plain}} \in \mathbb{R}^{m \times (n-k_i)}$ also varies across the clients, which makes the traditional aggregation methods of weight-averaging $\Delta\mathbf{W}_i^{\text{plain}}$ or $\mathbf{A}_i^{\text{plain}}$ inapplicable. Considering that the unencrypted columns of all the clients have been clustered to the left as explained in Section 3.2, we let the server apply column-wise weighted averaging to aggregate the unencrypted model parameters as illustrated in the top of Fig. 6. This column-wise partial aggregation is consistent with standard FedAvg (McMahan et al., 2017) under client subsampling and ensures no bias is introduced from non-contributing clients. The detailed algorithm is elaborated in Appendix D.2.

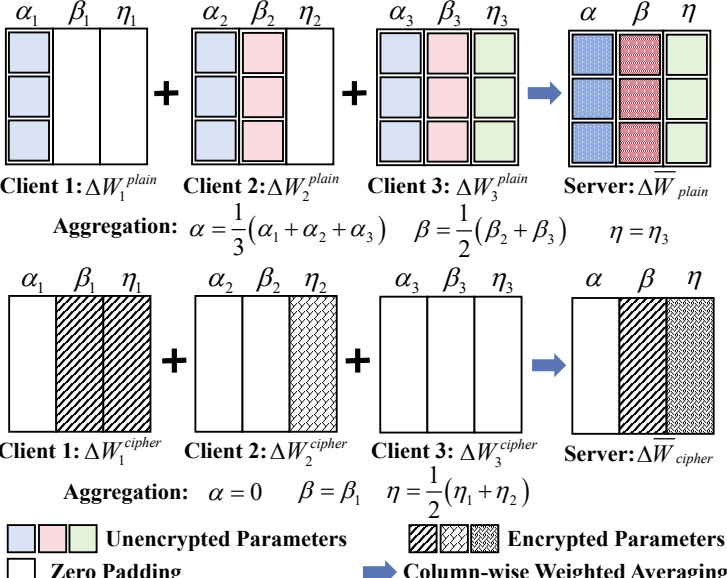

Figure 6: Aggregation of unencrypted (top) and encrypted (bottom) model parameters.

**Aggregation of Encrypted Model Parameters:** Similarly, upon receiving $\mathbf{B}_i$ and the encrypted part of $\mathbf{A}_i$ (denoted as $\mathbf{A}_i^{\text{cipher}} = \{C_j\}^{N^b}$) from Client $i$, the server calculates the encrypted weight update from Client $i$ as $\Delta\mathbf{W}_i^{\text{cipher}} = \mathbf{B}_i\mathbf{A}_i^{\text{cipher}} \in \mathbb{R}^{m \times k_i}$. Although the difference in $k_i$ also causes $\Delta\mathbf{W}_i^{\text{cipher}}$ to have different shapes, the cipher block order is consistent across the clients, and the encrypted columns of all the clients have been clustered to the right as explained in Section 3.2. Thus, the encrypted model parameters can be similarly aggregated via column-wise weighted averaging as illustrated in the bottom of Fig. 6. The detailed algorithm is elaborated in Appendix D.3.

To reduce communication overhead, the server sends the unencrypted and encrypted parts separately:

• For the unencrypted part, since most model parameters remain unencrypted during LoRA, we let the server, which generally has more computation capability, apply singular value decomposition (SVD) to decompose the aggregation result: $\Delta\overline{\mathbf{W}}_{\text{plain}} \overset{\text{SVD}}{=} \mathbf{U_P}\boldsymbol{\Sigma_P}\mathbf{V_P}^\top \in \mathbb{R}^{m \times K}$, where $K = n - \min(k_i)$, $\mathbf{U_P} \in \mathbb{R}^{m \times m}$, $\boldsymbol{\Sigma_P} \in \mathbb{R}^{m \times K}$ and $\mathbf{V_P}^\top \in \mathbb{R}^{K \times K}$. Then, given each client's rank $r_i$, the server slices

the decomposition results as: $\mathbf{U_p}[:, : r_i] \in \mathbb{R}^{m \times r_i}$, $\mathbf{\Sigma_p}[: r_i, : r_i] \in \mathbb{R}^{r_i \times r_i}$ and $\mathbf{V_p^\top}[: r_i, :] \in \mathbb{R}^{r_i \times K}$. Thus, each client receives the unencrypted aggregation result corresponding to its own rank.
• For the encrypted part, since each client only selectively encrypts a small portion of model parameters, we let the server first truncate the aggregation result by the encryption budget of respective clients, and then let the clients decrypt their corresponding aggregation results as in Section 3.4.

## 3.4 REPARAMETERIZATION

To enable the next round of model tuning, each client needs to merge the plaintext and ciphertext results returned by the server into new LoRA parameters. Suppose the client receives the plaintext decomposition result: $\Delta \overline{\mathbf{W}}_{\text{plain}} \stackrel{\text{SVD}}{=} \mathbf{B_p A_p}$, where $\mathbf{B_p} = \mathbf{U_p} \sqrt{\mathbf{\Sigma_p}} \in \mathbb{R}^{m \times r}$ and $\mathbf{A_p} = \sqrt{\mathbf{\Sigma_p}} \mathbf{V_p^\top} \in \mathbb{R}^{r \times K}$, and the cipher blocks from the server. For the plaintext decomposition results, the client directly zero-pads $\mathbf{B_p}$ and $\mathbf{A_p}$ to the shapes of $(m, r)$ and $(r, n)$, respectively, for further update of its model parameter matrix. For the cipher blocks, the client first decrypts the ciphertext into plaintext, and then performs SVD and zero-padding to obtain the decomposed results as $\Delta \overline{\mathbf{W}}_{\text{cipher}} \stackrel{\text{SVD}}{=} \mathbf{B_c A_c}$, where $\mathbf{B_c} = \mathbf{U_c} \sqrt{\mathbf{\Sigma_c}}$ and $\mathbf{A_c} = \sqrt{\mathbf{\Sigma_c}} \mathbf{V_c^\top}$. By merging the decomposition results of the plaintext and ciphertext via Eq. (4) and restoring the position of the model parameters according to the corresponding positions in Section 3.2, the correct aggregation result can be reparameterized as:

$$\mathbf{B_g} = [\mathbf{B_p}\ \mathbf{B_c}] = [\mathbf{U_p}\sqrt{\mathbf{\Sigma_p}}\ \mathbf{U_c}\sqrt{\mathbf{\Sigma_c}}] \in \mathbb{R}^{m \times (r+r)}, \mathbf{A_g} = \begin{bmatrix} \mathbf{A_P} \\ \mathbf{A_c} \end{bmatrix} = \begin{bmatrix} \sqrt{\mathbf{\Sigma_p}}\mathbf{V_P^\top} \\ \sqrt{\mathbf{\Sigma_c}}\mathbf{V_c^\top} \end{bmatrix} \in \mathbb{R}^{(r+r) \times n}. \quad (4)$$

Finally, the client performs SVD on $\mathbf{A_g}$ and $\mathbf{B_g}$ again, and re-adjusts the parameter shapes according to the client's rank to $\hat{\mathbf{B}} \in \mathbb{R}^{m \times r}$ and $\hat{\mathbf{A}} \in \mathbb{R}^{r \times n}$. As the SVD conducted by the client is all on low-rank matrices, the computation overhead is trivial. The detailed derivation of the reparameterization is presented in Appendix D.4. The formal proof of the losslessness of meaningful model updates in SHE-LoRA is provided in Appendix D.5.

## 3.5 PRIVACY GUARANTEE OF SHE-LoRA

The parameters protected by HE do not leak privacy, and privacy risks are mainly caused by un-encrypted parameters. The column swapping process, which serves as column-wise obfuscation, increases the difficulty of privacy attacks, and can be viewed as adding an asymptotic Gaussian-distributed noise (with $s^2$ as the equivalent variance) to the original gradient as detailed in Appendix D.7.

Thus, the privacy guarantee of selective encryption can be given by the Bayesian Cramér-Rao Lower Bound (Chen et al., 2025; Huang et al., 2025). Specifically, the reconstruction error $E_{\mathcal{R}}$ can be defined as the minimum expected squared reconstruction error:

$$E_{\mathcal{R}} = \min_{\mathcal{R}} \mathbb{E}_{x \sim \mathcal{X}} \mathbb{E}_{\mathbf{y} \sim f(g(x))} \left[ \left\| \mathcal{R}(\mathbf{y}) - x \right\|_2^2 \right] \quad (5)$$

where $\mathcal{R}(\cdot)$ is any data reconstruction attack method, $f(\cdot)$ is the selective HE method of SHE-LoRA, and $g(x)$ is the gradient calculated on data $x$. Then, the Bayesian Cramér-Rao Lower Bound can be given by (Huang et al., 2025):

$$E_{\mathcal{R}} \geq \frac{d^2}{\mathbb{E}_{x \sim \mathcal{X}}[\text{tr}(J_F(x))] + \lambda_e(J_P)} \geq \frac{d^2}{\frac{n(1-\gamma)}{s^2}\mathbb{E}_{x \sim \mathcal{X}}\|\nabla_x g(x)\|_{\max}^2 + \lambda_e(J_P)}, \quad (6)$$

where $J_F(x)$ is the Fisher information matrix, $\text{tr}(\cdot)$ is the trace operator, $\lambda_e(J_P)$ is the largest eigenvalue of the prior-informed Fisher information matrix $J_P$, $d$ is the data dimension, $n$ is the number of columns in $\mathbf{G}$, $\gamma$ is the encryption ratio, $s^2$ is the variance of the equivalent noise, and $\|\nabla_x g(x)\|_{\max}^2 = (\max_{i,j} |\nabla_x g_j(x_i)|)^2$ quantifies the maximum gradient exposure, which is defined as the squared maximum sensitivity of an unencrypted (exposed) gradient coordinate $g_j(x_i)$ with respect to a data feature $x_i$.

Eq. (6) means that the reconstruction error $E_{\mathcal{R}}$ is lower-bounded by a quantity whose denominator depends on $\text{tr}(J_F(x))$ and $\lambda_e(J_P)$. Since $\lambda_e(J_P)$ is determined solely by the data prior, the bound under fixed $n$, $s^2$, and $\gamma$ is governed exclusively by $\|\nabla_x g(x)\|_{\max}^2$. By selectively encrypting the most sensitive columns, which are measured in terms of the Wanda parameter sensitivity (a proxy for their contribution to input-space sensitivity via the score $S_j$ in Line 217), SHE-LoRA suppresses the dominant terms in the Fisher information matrix, thereby reducing the magnitude of the unencrypted gradients $\|\nabla_x g(x)\|_{\max}^2$ and lowering $\text{tr}(J_F(x))$.

In summary, by encrypting the most sensitive parameters, SHE-LoRA increases the minimum achievable reconstruction error and strengthens privacy against any gradient inversion attack.

## 4 PERFORMANCE EVALUATION

### 4.1 EXPERIMENTAL SETUP

**Model:** We select the Bert-Large (Devlin et al., 2018) model and the OpenLLaMA-3B (Geng & Liu, 2023) model for performance evaluation across diverse task scenarios. More evaluation settings and results on larger LLMs can be found in Appendix E.

Table 1: Heterogeneous device types.

| Type | GFlops | Rank | Budget (Bert, LLaMA) | # |
|------|--------|------|----------------------|---|
| 1 | 105.2 | 8 | (0.4%, 0.125%) | 20 |
| 2 | 165.5 | 16 | (0.4%, 0.125%) | 15 |
| 3 | 216.9 | 16 | (0.8%, 0.25%) | 10 |
| 4 | 243.1 | 32 | (1.6%, 0.50%) | 5 |

**Datasets:** We use the IMDB (Maas et al., 2011) and natural-instructions datasets (Wang et al., 2022) for natural language understanding and generation tasks, respectively. We use the MMLU (Hendrycks et al., 2021) and the GLUE (Wang et al., 2019) benchmarks for evaluation on natural language generation and natural language understanding tasks, respectively. For evaluation on vision tasks, we use the MNIST (LeCun et al., 2002), DTD (Cimpoi et al., 2014), EuroSAT (Helber et al., 2019), GTSRB (Stallkamp et al., 2012), SVHN (Netzer et al., 2011) visual classification datasets.

**Hyperparameters of HE:** We adopt the CKKS implementation from the TenSEAL library (Benaissa et al., 2021) for HE operations. As instructed[1], we set the polynomial degree to 8192, 2048, and the modules chain to [60, 40, 60], [20,20] for OpenLLaMA-3B and Bert-Large, respectively.

**Implementation:** SHE-LoRA is implemented with PyTorch based on the Flower framework[2]. We deploy federated LoRA of LLMs on 50 clients for 200 rounds. The heterogeneous data partitioning is instantiated via a Dirichlet distribution with $\rho = 0.3$. The evaluation settings and results on more clients can be found in Appendix E.1. As detailed in Table 1, we configure four types of client devices with varying computing capabilities, LoRA ranks and encryption budgets. Without losing generality, we posit that weaker devices are characterized by lower ranks and encryption budgets, while stronger devices are capable of supporting higher ranks and encryption budgets.

### 4.2 MODEL TUNING PERFORMANCE

We compare the model tuning performance of SHE-LoRA with two homogeneous LoRA-based methods (FedIT (Zhang et al., 2024) and FedSA (Guo et al., 2024)) and two heterogeneous LoRA-based methods (HeterLoRA (Cho et al., 2024) and Flex-LoRA (Bai et al., 2024)). The methods' performance on natural language understanding tasks and natural language generation tasks is evaluated on the GLUE benchmark (Wang et al., 2019) and the MMLU benchmark (Hendrycks et al., 2021), respectively, while the methods' performance on vision tasks is evaluated on the 5 visual classification datasets. SHE-LoRA achieves comparable performance to the SOTA method (Flex-LoRA) and outperforms the other baselines. Detailed results are elaborated in Appendix F.1.

### 4.3 HE COST EFFICIENCY

To our best knowledge, SHE-LoRA is the first to integrate SHE into federated LoRA of LLMs. We implement two methods for the comparison of HE cost efficiency: (1) **MaskCrypt** (Hu & Li, 2024), the SOTA SHE method for securing FL, and (2) **Baseline**, the vanilla method with full HE of LoRA parameters. Specifically, MaskCrypt lets each client select an encryption mask and uses the union of the masks for global SHE of parameters during FL. Baseline uses the stock implementation of CKKS (Benaissa et al., 2021) to encrypt each LoRA parameter. The clients' device specifications follow Table 1. We collect the encryption time and communication overhead of all the clients under the methods per round during the federated tuning of the OpenLLaMA-3B and Bert-Large models. Fig. 7 and Fig. 8 show the collected results, where the bar represents the mean, and the lines extending upward and downward from the mean represent the maximum and minimum values, respectively.

**Encryption Time**: Baseline always consumes the longest encryption time as it encrypts each LoRA parameter. In comparison, MaskCrypt greatly shortens the encryption time on both models, which is primarily due to its selective encryption of partial parameters. However, the clients' encryption time in both Baseline and MaskCrypt severely fluctuates within [311s, 653s] and [1.556s, 104.63s] on OpenLLaMA-3B, and [12s, 60s] and [0.27s, 39.75s] on Bert-Large, respectively. This is because that these methods cannot deal with the inflation of the global HE mask caused by matrix multiplication

---

[1] https://github.com/OpenMined/TenSEAL
[2] https://flower.ai/

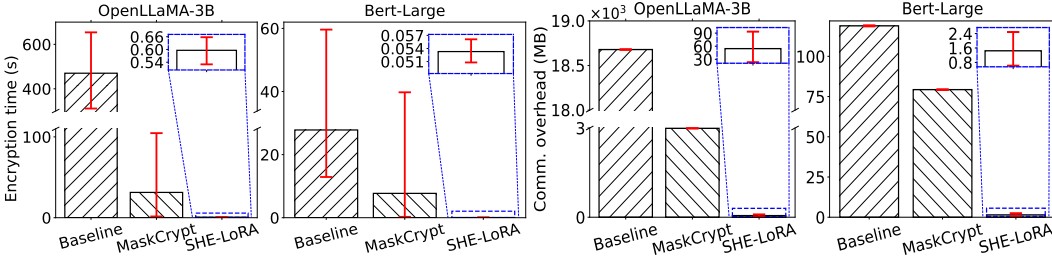

Figure 7: Encryption time.        Figure 8: Communication overhead.

and mask heterogeneity (Section 2.4). Thus, the clients in these methods, whether weak or strong, have to encrypt the same amount of parameters, causing highly imbalanced encryption time. In contrast, thanks to the column swapping and clustering of encrypted columns, which enables efficient utilization of CKKS key sizes, SHE-LoRA reduces the mean encryption time by 99.87%∼98.10% and 99.81%∼99.31% as compared to Baseline and MaskCrypt on the OpenLLaMA-3B and Bert-Large models, respectively. Moreover, we notice that even if the clients differ in computing capabilities, their encryption time under SHE-LoRA hardly fluctuates. The primary reason is that each client in SHE-LoRA can choose an affordable encryption budget $\gamma_i$ that matches its device capability, and hence results in no significant difference in encryption time across the clients.

**Communication Overhead**: The mean values under the methods follow similar rankings as those in encryption time, where SHE-LoRA reduces communication overhead by 99.71%∼98.18% on OpenLLaMA-3B and 98.78%∼98.18% on Bert-Large as compared to Baseline and MaskCrypt, respectively. The major difference lies in variations, where the communication overhead stays constant in Baseline and MaskCrypt, but fluctuates in SHE-LoRA. This is consistent with their designs: Baseline lets each client encrypt all the parameters (full ciphertext size) and MaskCrypt lets clients encrypt the global union of masks (partial ciphertext size), while SHE-LoRA lets clients choose encryption budgets that match their device capabilities (diverse ciphertext sizes).

## 4.4 RESISTANCE TO PRIVACY ATTACK

SHE-LoRA is applicable to both parameter and gradient updates. Although parameter-based attacks can be transformed into gradient-based ones, they are highly inaccurate and impractical for large models, even with direct access to gradients (Wang & Li, 2024). Therefore, we evaluate the resistance of SHE-LoRA against the DAGER attack (Petrov et al., 2024), which is the SOTA gradient inversion attack method. DAGER exploits the fact that gradients are linear combinations of input embeddings (Appendix B.2). It iterates over the entire vocabulary and measures the distance between each embedding vector and the principal components of the gradient, aiming to identify tokens. More results against membership inference attacks are detailed in Appendix F.4.

We conduct the federated LoRA of the OpenLLaMA-3B model via SHE-LoRA and MaskCrypt on two datasets, SST2 (Socher et al., 2013) and Rotten Tomatoes (Pang & Lee, 2005), as in DAGER, with $r$=256 and HE settings in Section 4.1. For fair comparison, we let them use the same HE overhead (ciphertext size). We also implement a non-private SOTA federated LoRA method, Flex-LoRA, and its privacy-preserving form with DP protection, Flex-LoRA-DP, where gradients are obfuscated with $\sigma^2$ noise, and $\sigma = 10^{-3}$ as in DAGER. We launch the DAGER attack on gradients per training round for each method over the two datasets, respectively, and collect the data reconstruction scores of DAGER under the batch sizes of $4, 8, 16$. The scores are collected in terms of "ROUGE-1" (R-1 in short), which measures the matching degree of unigrams, and "ROUGE-2" (R-2 in short), which measures the matching degree of bigrams. Smaller scores reflect better privacy protection.

Table 2 shows the mean and standard deviation of the scores. In practice, if SHE-LoRA and MaskCrypt use the encryption budget of the weakest device in Table 1 (i.e., $\gamma_i = 0.125\%$) for SHE, DAGER completely fails on both methods under all settings (i.e., scores=0). Thus, we gradually decrease $\gamma_i$ and find that DAGER begins to succeed in partially compromising SHE-LoRA when $\gamma_i < 0.3‰$, which consumes only one ciphertext packet. In contrast, MaskCrypt is no longer secure under the same HE overhead, while DP is far less secure than SHE-LoRA and may significantly degrade model accuracy (Sun et al., 2024c). The strong resistance of SHE-LoRA against DAGER is primarily due to the column swapping and SHE of important columns. Although the change in the principal components of gradients caused by column swapping is trivial, which does not lead to the failure of DAGER, such change leads to a strong perturbation in the orthogonal complement of gradients in the low-rank space of LoRA parameters, which causes the failure of DAGER's span

Table 2: Data reconstruction scores of DAGER.

| Dataset | Method | B=4 | | B=8 | | B=16 | |
|---|---|---|---|---|---|---|---|
| | | R-1 | R-2 | R-1 | R-2 | R-1 | R-2 |
| SST2 | Flex-LoRA | 95.18±1.6 | 94.66±1.8 | 61.14±1.9 | 52.49±2.2 | 10.27±1.6 | 5.86±1.2 |
| | Flex-LoRA-DP | 86.25±1.1 | 86.11±1.4 | 80.28±1.1 | 78.54±1.3 | 68.62±3.1 | 66.44±3.7 |
| | MaskCrypt | 89.16±1.3 | 87.93±2.1 | 61.49±2.2 | 61.49±2.4 | 10.91±1.2 | 6.79±1.4 |
| | SHE-LoRA | 0.72±5.2 | 0.12±1.2 | 0.98±4.4 | 0.14±0.6 | 0.0±0.0 | 0.0±0.0 |
| Rotten Tomatoes | Flex-LoRA | 38.44±1.5 | 32.76±1.3 | 3.76±1.4 | 2.12±2.1 | 0.0±0.0 | 0.0±0.0 |
| | Flex-LoRA-DP | 36.74±1.9 | 31.28±2.6 | 3.76±1.3 | 2.02±2.3 | 0.0±0.0 | 0.0±0.0 |
| | MaskCrypt | 31.65±2.0 | 25.11±2.6 | 6.09±1.0 | 3.27±1.2 | 0.0±0.0 | 0.0±0.0 |
| | SHE-LoRA | 0.0±0.0 | 0.0±0.0 | 0.0±0.0 | 0.0±0.0 | 0.0±0.0 | 0.0±0.0 |

check. In addition, as the key gradient information for reconstructing data has also been protected by SHE, DAGER completely fails when the batch size is greater than 8 even if only 0.3‰ parameters are encrypted.

We further decrease the batch size to 1 (i.e., easiest for inversion attack) with $\gamma_i = 0.3‰$ in SHE-LoRA, while gradually increasing the ratio=$\frac{\text{MaskCrypt HE overhead}}{\text{SHE-LoRA HE overhead}}$, and measure the data reconstruction scores of DAGER under the two methods, which are shown in Fig. 9. Due to ciphertext inflation, when $\frac{\text{MaskCrypt HE overhead}}{\text{SHE-LoRA HE overhead}}$=1, MaskCrypt is inefficient at protecting sufficient parameters against DAGER. To match SHE-LoRA's security, MaskCrypt has to consume $> 100\times$ the HE overhead of SHE-LoRA, making it unsuitable for weak clients.

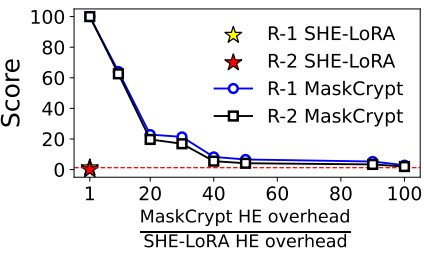

Figure 9: Resistance comparison.

### 4.5 PRIVACY LEAKAGE ANALYSIS FROM MUTUAL INFORMATION PERSPECTIVE

To delve into the root cause of SHE-LoRA's effectiveness, we perform a detailed analysis on how encryption budget and SHE strategy affect privacy leakage from the perspective of mutual information. Specifically, we implement three naive encryption strategies: (1) **Max:** the most important parameters are prioritized for encryption; (2) **Min:** the least important parameters are prioritized for encryption; (3) **Random:** parameters are randomly selected for encryption. Then, we gradually increase the encryption budget from 0.3‰ to 80%, and measure the mutual information shared between full parameters and the selectively

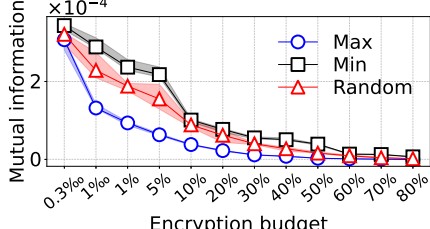

Figure 10: Impact of encryption budget and strategy on mutual information.

encrypted parameters per strategy via an efficient approach based on the kernel density estimators (Appendix F.3). Fig. 10 shows the measured results. We find that the mutual information in Max drops much faster than the others, while the mutual information in Min drops the slowest, reflecting a strong correlation between parameter importance and privacy leakage risk. The cost-effective protection of important parameters is the key to the effectiveness of SHE-LoRA against DAGER-like attacks.

## 5 CONCLUSION

We propose SHE-LoRA, a framework integrating SHE and LoRA for efficient and privacy-preserving federated tuning of LLMs in cross-device environments. It constrains the expansion of ciphertexts through HE subset negotiation, enables tailored privacy protection via selective encryption of parameters based on column-swapping parameter obfuscation, and achieves efficient and accurate update of LoRA parameters by column-aware adaptive aggregation and subsequent reparameterization. Results show that SHE-LoRA maintains model tuning performance comparable to non-private baselines, while achieving strong resistance to SOTA attacks, and significantly reducing communication overhead by 99.71% and encryption time by 99.87%, compared to HE baselines. Our work demonstrates that SHE with a well balance between privacy and utility can secure federated LoRA of LLMs against DAGER-like attacks and membership inference attacks. We hope SHE-LoRA can foster further research into creating more reliable and cost-effective privacy protection frameworks for private collaborative learning.

## 6 ACKNOWLEDGEMENTS

This work was supported in part by the National Key Research and Development Program of China under Grant 2022YFB4500800, in part by the National Natural Science Foundation of China under Grant 62103325, in part by the Shaanxi High-Level Talent Program under Grant 2021QCYRC4-26.

## 7 REPRODUCIBILITY STATEMENT

The paper has fully disclosed all the information needed to reproduce the main experimental results of the paper to the extent that it affects the main claims and conclusions of the paper. We have provided a link (https://github.com/liyan2015/SHE-LoRA) to a downloadable source code with detailed explanations and annotations to support the reproducibility of our work. In addition, the necessary hyperparameter settings are described in Section 4.1.

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

# Appendix

## LIST OF CONTENTS

## A    RELATED WORK

**LoRA Tuning in FL.** FedIT (Zhang et al., 2024) employed FedAvg to aggregate updates from locally performed LoRA fine-tuning. FeDeRA (Yan et al., 2024) modified the initialization of matrices **A** and **B** by the SVD results of pre-trained parameters, which helps mitigate client drift caused by data heterogeneity. SLoRA (Babakniya et al., 2023) employed SVD for initializing matrices **A** and **B**, and calculated a mask to reduce the parameters for training and communication overhead. FedRA (Su et al., 2024) partitioned the pre-trained model by layer, enabling clients to fine-tune specific layers on homogeneous models. Inspired by the mixture of experts architecture, HydraLoRA (Tian et al., 2024) proposed to learn multiple LoRAs corresponding to different knowledge. PiSSA (Meng et al., 2024) proposed to directly fine-tune the principal component of the pre-trained model, facilitating rapid convergence and enhancing overall performance. These LoRA variants primarily focus on reducing training costs in homogeneous settings, but neglect the heterogeneity of device capabilities and Non-IID data in cross-device federated PEFT scenarios.

**Heterogeneous LoRA.** FLoRA (Wang et al., 2024) contended that the aggregation used in FedIT (Zhang et al., 2024) is flawed and employed stacking to aggregate parameters from heterogeneous clients. HeterLoRA (Cho et al., 2024) implemented different ranks on clients and aggregates heterogeneous LoRA modules through zero-padding, which may dilute certain parameters. RBLA (Chen et al., 2024) designed a rank-based LoRA aggregation method to prevent model parameter dilution caused by zero-padding. Flex-LoRA (Bai et al., 2024) synthesized a complete set of LoRA weights from individual client contributions, and employed SVD for weight reparameterization, thereby fully leveraging heterogeneous client resources. Furthermore, pFedLoRA (Yi et al., 2023) aggregated adapters to facilitate personalized FL, and treated LoRA as a mechanism for knowledge transfer, allowing clients to locally train heterogeneous models while maintaining a homogeneous adapter. Although these works have made significant contributions to heterogeneous federated PEFT with LoRA, they still suffer from potential privacy leakage risks under inversion attacks.

**Privacy Preservation Techniques.** To defend against inversion attacks, Yu et al. (2022) employed the DP-SGD optimizer for fine-tuning, providing formal DP guarantees for model parameters. Considering that DP noise may be amplified by LoRA multiplication, FFA-LORA (Sun et al., 2024c) modified the LoRA training procedure by freezing matrix **A** after initialization and solely applying DP on matrix **B**, at the cost of fewer tunable model parameters. Similarly, FedSA-LoRA (Guo et al., 2024) proposed to globally train matrix **A** while reserving matrix **B** for local training without participating in aggregation. Han & Yan (2023) proposed an adaptive, precision-lossless batch HE method that transforms model parameters into non-negative values to prevent overflow errors. Inspired by model pruning techniques, FedML-HE (Jin et al., 2023) proposed to encrypt only a subset of sensitive model parameters to reduce HE overhead. MaskCrypt (Hu & Li, 2024) proposed to select a consensus mask for SHE to minimize overhead across homogeneous devices. In summary, the application of DP involves a trade-off between privacy protection and model convergence, while existing HE methods struggle to balance privacy and efficiency in cross-device federated PEFT with LoRA, particularly under scenarios with Non-IID data and heterogeneous device capabilities.

## B    PRELIMINARY

### B.1    RELATIONSHIP BETWEEN PARAMETER IMPORTANCE AND PRIVACY LEAKAGE RISK

On one hand, as indicated in previous SHE methods (Hu & Li, 2024; Jin et al., 2023), the privacy leakage risk in FL mainly comes from the fact that the locally trained model weights contain private data information, and are vulnerable to attacks. The lower gradient loss a parameter results in, the higher privacy leakage risk it may cause under the attacks.

On the other hand, model pruning methods (Frantar et al., 2023; Hassibi et al., 1993; Sun et al., 2024b) have proved that by carefully screening the parameters by importance, many parameters can be removed without hurting performance. Following this rationale, SHE-LoRA lets heterogeneous clients adaptively encrypt partial parameters via importance screening and negotiate a global HE subset for secure Federated PEFT without hurting privacy and efficiency.

Specifically, let **W** and $\mathcal{L}(\mathbf{W})$ denote parameters and the loss function, respectively. Given a subset of the parameters $\mathbf{w} \in \mathbf{W}$, $\mathbf{W}_{-\mathbf{w}} = \mathbf{W} - \mathbf{w}$ denotes the parameters with $\mathbf{w}$ zeroed out. According to (Hassibi et al., 1993), the loss function $\mathcal{L}(\mathbf{W}_{-\mathbf{w}})$ can be expanded as the following Taylor series:

$$\mathcal{L}(\mathbf{W}_{-\mathbf{w}}) = \mathcal{L}(\mathbf{W} - \mathbf{w}) = \mathcal{L}(\mathbf{W}) + g^{\top}\mathbf{w} + \frac{1}{2}\mathbf{w}^{\top}H\mathbf{w} + O(\|\mathbf{w}\|^3), \tag{7}$$

where $g^{\top}$ and $H$ are the first-order (gradient) and second-order (Hessian) partial derivatives, respectively. $O(\|\mathbf{w}\|^3)$ is the higher-order infinitesimal of $\mathbf{w}$. The sensitivity of $\mathbf{w}$ (denoted as $\Omega(\mathbf{w})$) can be denoted as:

$$\Omega(\mathbf{w}) = |\mathcal{L}(\mathbf{W}_{-\mathbf{w}}) - \mathcal{L}(\mathbf{W})| = g^{\top}\mathbf{w} + \frac{1}{2}\mathbf{w}^{\top}H\mathbf{w} + O(\|\mathbf{w}\|^3) \approx \frac{1}{2}\mathbf{w}^{\top}H\mathbf{w}. \tag{8}$$

Considering that when $\mathbf{W}$ converges to the local minimum after training, $g^{\top}$ is 0, while $O(\|\mathbf{w}\|^3)$ can be neglected, thus $\Omega(\mathbf{w})$ only depends on $H$ and $\mathbf{w}$. Finally, the sensitivity of the $q$-th parameter is calculated as $\Omega(\mathbf{w}_q) = \frac{\mathbf{w}_q^2}{2[H^{-1}]_{qq}}$ according to OBS (Hassibi et al., 1993), where $[H^{-1}]_{qq}$ is the diagonal element at $(q, q)$ of the inverse Hessian matrix, $H^{-1}$.

Inspired by $H = XX^{\top}$ from SparseGPT (Frantar & Alistarh, 2023), Wanda (Sun et al., 2024b) modified weight importance assessment to avoid the high computation cost of $H$ and $H^{-1}$ (Eq. (2)). Based on Wanda, SHE-LoRA assesses channel-wise weight importance as in Section 3.1.1.

Besides, Fig. 10 shows that the mutual information (privacy leakage) of *Max* decreases much sharper than the others along with the encryption of parameters, which intuitively reflects a strong correlation between weight importance and privacy leakage risk.

## B.2 DATA RECONSTRUCTION VIA INVERSION ATTACKS

Inversion attacks (Petrov et al., 2024; Balunovic et al., 2022; Zhu et al., 2019; Jeon et al., 2021) aim to reconstruct private training data or model parameters, such as pixel values in images or sensitive information in text, by reversing the clients' updates uploaded during federated fine-tuning, such as gradients, parameter updates or prediction results. Specifically, for a model $f_{\mathbf{W}}(x) = \mathbf{W}^{\top}x$, a client trains it with its local data $(x, y)$, and calculates the gradient $g$ as the derivative of the loss function $\mathcal{L}$:

$$g = \frac{\partial \mathcal{L}}{\partial \mathbf{W}} = \frac{\partial \mathcal{L}}{\partial f_{\mathbf{W}}} \frac{\partial f_{\mathbf{W}}}{\partial \mathbf{W}} = \frac{\partial \mathcal{L}}{\partial f_{\mathbf{W}}} \cdot x, \tag{9}$$

and uploads it to the server. The uploaded gradient $g$ contains a linear combination of the original data. An attacker can analyze the principal components of the model update to search for the client's data distribution space and then reconstruct the data (Petrov et al., 2024).

## B.3 FEDERATED PARAMETER-EFFICIENT FINE-TUNING

Federated Parameter-Efficient Fine-Tuning (PEFT) is a technology designed for the efficient fine-tuning of large models within a distributed learning framework that prioritizes data privacy. For instance, LoRA (Hu et al., 2022) is a well-established method for PEFT. Suppose the weight matrix of the global pre-trained model is denoted as $\mathbf{W} \in \mathbb{R}^{m \times n}$, LoRA introduces trainable parameters, $\mathbf{B} \in \mathbb{R}^{m \times r}$ and $\mathbf{A} \in \mathbb{R}^{r \times n}$. Each client freezes the original weights $\mathbf{W}$ and learns only the trainable low-rank parameters $\mathbf{B}$ and $\mathbf{A}$ on its local data. Then, the updates (e.g., weights or gradients) from clients are aggregated by the server via

$$\Delta \overline{\mathbf{W}} = \sum_{i=1}^{N} \tau_i \mathbf{B_i} \mathbf{A_i}, \tag{10}$$

where $\tau_i$ denotes the weight coefficient of Client $i$, proportional to its local data size. Federated PEFT uses Eq. (10) to guarantee heterogeneous aggregation of LoRA parameters. SVD is used to reparameterize the aggregation result into heterogeneous LoRA parameters, which reduces the communication overhead of $\Delta \overline{W}$. The new LoRA parameters are then sent back to each device for continued local training, and the process is iterated repeatedly to progressively optimize the model. It is worth noting that the model parameters are transmitted in plaintext throughout this process, making them visible to the server.

## B.4 MATRIX MULTIPLICATION IN LORA AMPLIFIES DP NOISE

Differential privacy (DP) (Yu et al., 2022; Zhu et al., 2025) is a common privacy defense mechanism that adds specific noise to gradient or parameter updates, making the relationship between gradients and data non-linear, which misleads the attacker's optimization direction. Given that the LoRA

fine-tuning parameter is $\Delta \mathbf{W} = \mathbf{BA}$, and $\Delta \mathbf{W}$ plays a role during model inference, if a noise $\epsilon$ satisfying the privacy budget is added to the parameters, then:

$$\Delta \mathbf{W} = (\mathbf{B} + \epsilon_B)(\mathbf{A} + \epsilon_A) = \mathbf{BA} + \mathbf{B}\epsilon_A + \mathbf{A}\epsilon_B + \epsilon_A \epsilon_B \qquad (11)$$

Thus, except for $\mathbf{BA}$, all the terms in Eq. (11) are noises that will be aggregated into the global model parameters, which may severely affect the model's performance and convergence direction.

### B.5 HOMOMORPHIC ENCRYPTION

Homomorphic encryption (Rivest et al., 1978; Gentry, 2009) is a cryptographic primitive that allows computations to be performed on encrypted data without revealing the underlying plaintext. It is exceptionally well-suited for FL, as it enables the computation of aggregation in server without exposing clients' updates. The Cheon–Kim–Kim–Song (CKKS) encryption scheme (Cheon et al., 2017) supports approximate numerical computations and floating-point arithmetic. The CKKS offers relatively high computational efficiency and supports vectorized operations, making it highly suitable for LoRA tuning in FL. It can be used to encrypt the parameters or gradients of local models, allowing the server to perform model aggregation without decrypting any data, thereby protecting user privacy. In the implementation of SHE-LoRA, we employ the CKKS provided by the TenSEAL library, which supports homomorphic computations on both vectors and tensors, thereby enabling encrypted computation operations over complex model parameters.

### B.6 ORDER-PRESERVING ENCRYPTION

Order-Preserving Encryption (OPE) (Agrawal et al., 2004; Boldyreva et al., 2009) is a cryptographic technique that preserves the numerical order of plaintexts. If two plaintexts, $a$ and $b$, satisfy the condition $a < b$, then their encrypted ciphertexts will also satisfy $Enc(a) < Enc(b)$. This enables comparisons, sorting, or range queries to be performed on ciphertexts without decryption. In principle, OPE maps plaintexts to an interval within the ciphertext space, while ensuring that the mapping function is monotonically increasing. Although OPE does not provide semantic security in the traditional sense (i.e., it is possible to infer the approximate range of the plaintext from the ciphertext), it is highly useful in applications that require sorting or range operations on encrypted data, such as encrypted databases or cloud storage queries. In SHE-LoRA, we employ OPE to hide clients' HE subset, which prevents the server from snooping on the positions of important model parameters and conducting targeted attacks.

### B.7 SINGULAR VALUE DECOMPOSITION

Singular value decomposition (SVD) (Klema & Laub, 1980) is a mathematical method that decomposes any real or complex matrix into the product of three standard matrices. It is applicable to matrices of arbitrary shapes, and hence suitable for tasks such as dimensionality reduction, data compression, and recommendation systems. For a real matrix $\mathbf{W} \in \mathbb{R}^{m \times n}$, SVD decomposes it as follows: $\mathbf{W} = \mathbf{U}\mathbf{\Sigma}\mathbf{V}^\top$, where $\mathbf{U} \in \mathbb{R}^{m \times m}$ is an orthogonal matrix, $\mathbf{\Sigma} \in \mathbb{R}^{m \times n}$ is a diagonal matrix with non-negative real values on its diagonal, known as singular values, arranged in descending order, and $\mathbf{V}^\top \in \mathbb{R}^{n \times n}$ is an orthogonal matrix as well. To match heterogeneous LoRAs from clients, many methods (Yan et al., 2024; Babakniya et al., 2023) have employed SVD decomposition, breaking down the aggregated matrix into $\mathbf{B} = \mathbf{U}\sqrt{\mathbf{\Sigma}}$ and $\mathbf{A} = \sqrt{\mathbf{\Sigma}}\mathbf{V}^\top$, which enables the projection of the original parameters into a lower-dimensional space while preserving the essential features.

## C  HE KEY MANAGEMENT AND DISTRIBUTION

In the configuration outlined herein, the system necessitates an honest-but-curious server to execute the aggregation of models. Then, a trusted third party is required to oversee the distribution of keys for both OPE and HE, with the assumption that it will not collude with the server. However, when the server colludes with compromised clients, cryptographic techniques such as threshold HE, Multi-Key HE, and proxy re-encryption can be employed to achieve distributed secure computation. **Threshold Homomorphic Encryption** (Aloufi et al., 2021) is a cryptographic scheme that amalgamates the threshold cryptography and HE. It not only supports computations on encrypted data (homomorphism), but also enables joint decryption by multiple participants (thresholding), thereby enhancing

both security and fault tolerance while preserving privacy. Decryption can only be accomplished with the collective participation of a predetermined number of participants, known as the threshold. **Proxy Re-Encryption** (Blaze et al., 1998; Ateniese et al., 2006) permits a semi-trusted third party (the proxy) to transform ciphertext encrypted for one party into ciphertext decryptable by another, all without accessing the original plaintext or either party's private keys. **Multi-Key Homomorphic Encryption** (Aloufi et al., 2021) enables homomorphic operations to be performed on user data encrypted under different keys, obviating the need for key sharing. The resultant ciphertext requires collaborative partial decryption by all participants, each utilizing their respective private key, to yield the final plaintext outcome. This mechanism ensures that joint computations can be performed in multi-party data collaborations while protecting the data privacy of each participant.

In SHE-LoRA, we retain Multi-Key HE as a countermeasure against potential collusion between the server and clients, while leaving its practical integration and optimization to future endeavors.

## D  ADDITIONAL TECHNICAL DETAILS OF SHE-LoRA

### D.1  THE ALGORITHM OF HE SUBSET NEGOTIATION

#### D.1.1  OBJECTIVES OF THE NEGOTIATION

The negotiation aims to select a subset of columns $Res \subseteq \bigcup_{i=1}^{N} G_i$ with cardinality $|Res| = \max_{i \in \{N\}} k_i$, where $k_i$ denotes the number of sensitive columns that client $i$ can afford to encrypt and $\{N\}$ denotes the set of all clients, thereby achieving a principled trade-off between privacy and HE overhead. However, the notions of "privacy" and "HE overhead" are inherently ambiguous without concrete metrics. To make the trade-off operational, we formalize two explicit objectives with *min-Coverage* and *max-Risk*.

We first define the $Coverage_i$ of client $i$ as the fraction of its sensitive columns that are selected for encryption (i.e., included in *Res*), and hence protected from exposure: $Coverage_i = \frac{|Res \cap G_i|}{|G_i|}$, where $G_i$ is the set of columns that client $i$ deems sensitive, and *Res* is the final selected subset.

To extend HE-based privacy protection to as many clients as possible, we adopt a max-min criterion and define the overall coverage as:

$$min\text{-}Coverage = \min_{i \in \{N\}} Coverage_i. \tag{12}$$

Maximizing *min-Coverage* ensures that the coverage of every client is at least this value, thereby guaranteeing the worst-case coverage of sensitive parameter columns: even the least-covered client receives a quantifiable level of protection.

Second, to quantify the privacy leakage risk of each client, we define $Risk_i = \frac{\sum_{j \in G_i \setminus Res} S_j}{\sum_{j \in G_i} S_j}$, which measures the fraction of client $i$'s total sensitivity that remains unencrypted. We further define

$$max\text{-}Risk = \max_{i \in \{N\}} Risk_i \tag{13}$$

to capture the worst-case privacy leakage risk across all clients, which we aim to minimize. Lower residual risk implies stronger privacy preservation. Minimizing *max-Risk* ensures that the privacy leakage risk of every client is at most this value, thereby providing a worst-case privacy guarantee: even the most exposed client suffers from no more than a quantifiable level of privacy leakage risk.

The negotiation thus seeks a balanced *Res* that jointly optimizes both Eq. (12) and Eq. (13). To formalize this goal, we define a composite objective score as Eq. (14):

$$\mathsf{score}(Res) = \underbrace{\min_{i \in \{N\}} \frac{|Res \cap G_i|}{|G_i|}}_{min\text{-}Coverage} - \underbrace{\max_{i \in \{N\}} \frac{\sum_{j \in G_i \setminus Res} S_j}{\sum_{j \in G_i} S_j}}_{max\text{-}Risk}. \tag{14}$$

Eq. (14) balances the minimal client coverage of sensitive parameters and the maximal privacy leakage risk of unencrypted parameters. These two objectives often contradict under a limited encryption budget: satisfying client-specific privacy leakage risk may sacrifice overall coverage of clients' sensitive parameters, and vice versa.

---

**Algorithm 1:** HE subset negotiation

**Input:** $clients = \{r_i, k_i, (G_i, S_i)\}_{i=1}^{N}$, $a, b, c$ are three hyper-parameters of selection ratio.
**Output:** $Res$: the column index of global HE subset.

1   $Res, selected\_num \leftarrow \{\}, 0$;
2   $Common \leftarrow$ sorts columns in $\bigcup_i G_i$ from most to least frequently deemed as sensitive;
3   $Sensitivity \leftarrow$ sorts columns in $\bigcup_i G_i$ from highest to lowest sensitivity;
4   $\Gamma \leftarrow \{(k, count_k)\}$ clusters clients by budget $k$, and sorts them in ascending order of $k$;
5   **for** *each budget $k$ in $\Gamma$* **do**
6      $\lambda \leftarrow$ update current budget by $k - selected\_num$;
7      $Clients \leftarrow$ collect all unique columns from $\bigcup_{i:k_i=k} G_i$, and sort them by $\min S_j$;
8      **if** $count_k = 1$ **then**
9          $P \leftarrow$ select top $\lambda$ columns from $G_i$-$Res$ of the unique client;
10          $Res \leftarrow \{P, \{Res\}\}$;
11          $selected\_num \leftarrow k$;
12      **else**
13          $a, b, c \leftarrow$ coefficients optimized via Bayesian optimization under $a + b + c = 1$, balancing *min-Coverage* and *max-Risk* as detailed in Algorithm 2;
14          $P \leftarrow$ select top $\lfloor a\lambda \rfloor$ columns from *Clients-Res*;
15          $C \leftarrow$ select top $\lfloor b\lambda \rfloor$ columns from *Common-P-Res*;
16          $S \leftarrow$ select top $\lambda - \lfloor a\lambda \rfloor - \lfloor b\lambda \rfloor$ columns from *Sensitivity-P-C-Res*;
17          $Res \leftarrow$ result of $k$ columns is $\{S, C, P, \{Res\}\}$;
18          $selected\_num \leftarrow k$;

19   **return** $Res$

---

### D.1.2   PROCEDURE OF THE NEGOTIATION

The workflow of HE subset negotiation is summarized as Algorithm 1. Note that the server keeps the rank $r_i$ and encryption budget $\gamma_i$ of all clients. The number of encrypted columns of Client $i$ is denoted as $k_i = \gamma_i \cdot n$, where $n$ is the number of columns in the parameter matrix. As described in Section 3.1.2, the server receives a set of tuples $(G_i, S_i)$ from clients as input, where $G_i$ is Client $i$'s set of columns that needs HE, and $S_i$ is their sensitivities.

At Lines 1-3, the server first initiates the negotiation result *Res* and the number of columns that have been selected *selected_num*. Then, the server maintains two shared lists: the *Common* list, which sorts all columns in $\bigcup_i G_i$ from most to least frequently deemed as sensitive, and the *Sensitivity* list, which sorts all columns in $\bigcup_i G_i$ from highest to lowest sensitivity. The columns ranked higher in *Common* are more frequently deemed as sensitive by the clients, and selecting them improves *min-Coverage*. The columns ranked higher in *Sensitivity* have greater global sensitivity, and selecting them reduces *max-Risk*. At Line 4, the server clusters the clients by their budget $k$, and sorts them in the ascending order of $k$. At Lines 5-18, the server repeats the process several times, which depends on the number of unique budgets. For each process, the number of column positions to be negotiated, denoted as $\lambda$, is calculated by the difference between current budget $k$ and the number of columns that have been determined. A *Clients* list ranks unique columns from budget-$k$ clients by their minimum sensitivity, aiming to encrypt columns that are personally deemed as sensitive. If the current budget corresponds to a single client (i.e., $count_k = 1$), the strategy greedily selects from that client's $G_i$ for optimal privacy protection. Otherwise, the server iteratively selects $\lfloor a\lambda \rfloor$ columns from *Clients*, $\lfloor b\lambda \rfloor$ columns from *Common*, and $\lambda - \lfloor a\lambda \rfloor - \lfloor b\lambda \rfloor$ columns from *Sensitivity* without duplicate selection to form the global HE subset, where $a + b + c = 1$ and $\lfloor \cdot \rfloor$ is the floor function. This hybrid selection jointly optimizes worst-case coverage and privacy risk while preserving a degree of client personalization to accommodate heterogeneous environments.

Clearly, the success of this objective relies heavily on the choice of the coefficients $a$, $b$, and $c$. As presented in Algorithm 2, we employ the Bayesian optimization (Swersky et al., 2013) to determine their optimal values via searching different combinations of column selection from three complementary perspectives: client-specific (*Clients*), commonly shared (*Common*), and sensitivity-driven (*Sensitivity*). Specifically, Algorithm 2 begins by defining the feasible search space for the parameters $(a, b, c)$ at Line 1, subject to the constraint $a + b + c = 1$. At Line 2, a Bayesian optimization model (i.e., a Gaussian process) is initialized over this search space. At Lines 3–16, the model performs $N_{\text{opt}}$ (e.g., 50) iterations of optimization: at each iteration $t$, the current parameter triple $(a_t, b_t, c_t)$ is used to construct the negotiation result $Res_t$ at Line 5, following the same column

---

**Algorithm 2:** Bayesian optimization for the selection of $a$, $b$ and $c$.

**Input:** $clients = \{r_i, k_i, (G_i, S_i)\}_{i=1}^N$, Current $\lambda$, *Res*, *Clients*, *Common*, *Sensitivity*.
**Output:** Optimal coefficients $(a^*, b^*, c^*)$ with $a^* + b^* + c^* = 1$

1   $\mathcal{X} \leftarrow$ Define search space with $\{(a, b) \in [0, 1]^2 \mid a + b \leq 1\}$;
2   Initialize a Bayesian optimization model $\mathcal{M}$ over $\mathcal{X}$;
3   **for** $t = 1$ **to** $N_{opt}$ **do**
     // Bayesian optimizer selects $(a_t, b_t)$ for evaluation
4     $(a_t, b_t) \leftarrow \text{SELECT}(\mathcal{M})$;
5     $c_t \leftarrow 1 - a_t - b_t$;
6     $P_t \leftarrow$ select top $\lfloor a_t \lambda \rfloor$ columns from *Clients-Res*;
7     $C_t \leftarrow$ select top $\lfloor b_t \lambda \rfloor$ columns from *Common-Res-$P_t$*;
8     $S_t \leftarrow$ select top $\lambda - \lfloor a_t \lambda \rfloor - \lfloor b_t \lambda \rfloor$ columns from *Sensitivity-Res-$P_t$-$C_t$*;
9     $Res_t \leftarrow$ the union of $\{S_t, C_t, P_t\}$;
10    **for** *each client $i$* **do**
11      $Coverage_i \leftarrow \dfrac{|Res_t \cap G_i|}{|G_i|}$;
12      $Risk_i = \dfrac{\sum_{j \in G_i \setminus Res} S_j}{\sum_{j \in G_i} S_j}$;
13    *min-Coverage* $\leftarrow \min_{i \in \{N\}} Coverage_i$;
14    *max-Risk* $\leftarrow \max_{i \in \{N\}} Risk_i$;
15    $\text{score}_t \leftarrow$ *min-Coverage* $-$ *max-Risk*;
     // Update Bayesian optimizer with $(a_t, b_t, \text{score}_t)$
16    $\mathcal{M} \leftarrow \text{UPDATE}(\mathcal{M}, (a_t, b_t), \text{score}_t)$;
17 $(a^*, b^*) \leftarrow \arg\max_t \text{score}_t$;
18 $c^* \leftarrow 1 - a^* - b^*$;
19 **return** $(a^*, b^*, c^*)$;

---

selection procedure as in Algorithm 1. Then, at Lines 10–12, the client-specific coverage and privacy risk are evaluated for every client. The overall score is computed at Line 15 using Eq. 14, and in Line 16, this score is fed back to update the Bayesian optimization model, guiding the next parameter selection. Finally, Lines 17–19 return the best-performing coefficients $(a^*, b^*, c^*)$ as the outcome of the negotiation.

### D.2   AGGREGATION OF UNENCRYPTED MODEL PARAMETERS

Algorithm 3 illustrates how the server aggregates unencrypted model parameters. The input of the algorithm is the set of unencrypted weight updates $\Delta W_i^{\text{plain}}$ from all $N$ clients. At Line 1, the columns of the aggregation result is initialized by $K = n - \min(k_1, \ldots, k_N)$, where $n$ is the number of columns in the frozen pre-trained parameter matrix, and $k_i$ is the number of encrypted columns of Client $i$. At Lines 2-3, the server initializes the aggregation result to $\mathbf{0} \in \mathbb{R}^{m \times K}$, and sets a counter that records the respective contributions of the clients during the aggregation of each column. At Lines 4-7, the server incorporates Client $i$'s parameters $\Delta W_i^{\text{plain}}$ into the aggregation results, and updates the counter to record the number of clients contributing to each column. At Lines 8-11, the server weight-averages the results based on the counters and returns the final aggregated result $\Delta \overline{W}_{\text{plain}} \in \mathbb{R}^{m \times K}$.

### D.3   AGGREGATION OF ENCRYPTED MODEL PARAMETERS

Algorithm 4 illustrates how the server aggregates encrypted model parameters. The algorithm takes the set of ciphertexts $\Delta W_i^{\text{cipher}} \in \mathbb{R}^{m \times k_i}$ as the input. At Line 1, the columns of the aggregation result are initialized by $K^* = \max(k_1, k_2, \ldots, k_N)$, where $k_i$ is the column number of encrypted model parameters of client $i$. At Lines 2-3, the server initializes the aggregation result to $\mathbf{0}_{1 \times K^*}$ and sets a counter to record the respective contributions of the clients. At Lines 4-6, the server incorporates encrypted model parameters $\Delta W_i^{\text{cipher}}$ into the aggregation results and updates the counter to record the number of clients contributing to each column. Finally, at Lines 7-9, the server weight-averages the encrypted model parameters based on the counters, and returns the final aggregated encrypted model parameters, denoted as $\Delta \overline{W}_{\text{cipher}} \in \mathbb{R}^{m \times K^*}$. Although the columns of encrypted model parameters extends to $K^*$, each client can receive a set of encrypted blocks matching its encryption budget.

---

**Algorithm 3:** Aggregation of unencrypted model parameters

---

**Input:** $\{\Delta W_i^{\text{plain}}\}_{i=1}^N$: Set of $N$ matrices, each $\Delta W_i^{\text{plain}}$ has shape $(m, n - k_i)$.
**Output:** $\Delta \overline{W}_{\text{plain}}$: Aggregated matrix with shape $(m, K)$.

1 $K \leftarrow n - \min(k_1, \ldots, k_N)$ ;
2 $\Delta \overline{W}_{\text{plain}} \leftarrow \mathbf{0}_{m \times K}$ ;
3 Counts $\leftarrow \mathbf{0}_{1 \times K}$ ;
4 **for** *each client* $i = 1$ **to** $N$ **do**
5 $\quad$ $c_i \leftarrow$ get column count of $\Delta W_i^{\text{plain}}$ ;
6 $\quad$ $\Delta \overline{W}_{\text{plain}}[:, : c_i] \leftarrow \Delta \overline{W}_{\text{plain}}[:, : c_i] + \Delta W_i^{\text{plain}}$;
7 $\quad$ Counts$[: c_i] \leftarrow$ Counts$[: c_i] + 1$ ;
8 **for** $j = 1$ **to** $K$ **do**
9 $\quad$ **if** Counts$[j] > 0$ **then**
10 $\quad\quad$ $\Delta \overline{W}_{\text{plain}}[:, j] \leftarrow \Delta \overline{W}_{\text{plain}}[:, j]/$Counts$[j]$;

11 **return** $\Delta \overline{W}_{plain}$

---

**Algorithm 4:** Aggregation of encrypted model parameters

---

**Input:** $\{\Delta W_i^{\text{cipher}}\}_{i=1}^N$: Sets of $N$ ciphertexts from clients, each $\Delta W_i^{\text{cipher}}$ has shape $(m, k_i)$.
**Output:** $\Delta \overline{W}_{\text{cipher}}$: Aggregated matrix with shape $(m, K^*)$.

1 $K^* \leftarrow \max(k_1, k_2, \ldots, k_N)$ ;
2 $\Delta \overline{W}_{\text{cipher}} \leftarrow \mathbf{0}_{m \times K^*}$ ;
3 Counts $\leftarrow \mathbf{0}_{1 \times K^*}$ ;
4 **for** *each client* $i = 1$ **to** $N$ **do**
5 $\quad$ $\Delta \overline{W}_{\text{cipher}}[:, -k_i :] \leftarrow \Delta \overline{W}_{\text{cipher}}[:, -k_i :] + \Delta W_i^{\text{cipher}}$
6 $\quad$ Counts$[-k_i :] \leftarrow$ Counts$[-k_i :] + 1$ ;
7 **for** $j = 1$ **to** $K^*$ **do**
8 $\quad$ **if** Counts$[j] > 0$ **then**
9 $\quad\quad$ $\Delta \overline{W}_{\text{cipher}}[:, j] \leftarrow \Delta \overline{W}_{\text{cipher}}[:, j]/$Counts[j]

10 **return** $\Delta \overline{W}_{cipher}$

---

### D.4 REPARAMETERIZATION OF LoRA

The updated full-parameter for each client, termed as $\Delta \mathbf{W}$, can be formulated as two parts, the plaintext update $\Delta \overline{\mathbf{W}}_{\text{plain}} = \mathbf{B}_p \mathbf{A}_p$ and the ciphertext update $\Delta \overline{\mathbf{W}}_{\text{cipher}} \in \mathbb{R}^{r \times k_i}$ as shown in Eq. (15). In order to reparameterize the two parts of the model parameters into the parameter matrices $\hat{\mathbf{B}}$ and $\hat{\mathbf{A}}$ of LoRA, we first apply SVD and zero-padding to the ciphertext update to generate two low-rank matrices ($\mathbf{B}_c \in \mathbb{R}^{m \times r}, \mathbf{A}_c \in \mathbb{R}^{r \times n}$), which ensures their dimension aligns with $\mathbf{B}_p \mathbf{A}_p$. The final LoRA parameter matrices $(\hat{\mathbf{B}}, \hat{\mathbf{A}})$ are calculated as follows:

$$
\begin{aligned}
\Delta \mathbf{W} &= \Delta \overline{\mathbf{W}}_{\text{plain}} + \Delta \overline{\mathbf{W}}_{\text{cipher}} \\
&\stackrel{\text{SVD}}{=\!=\!=} \mathbf{B}_p \mathbf{A}_p + \mathbf{B}_c \mathbf{A}_c \\
&= (\mathbf{U}_1 \sqrt{\mathbf{\Sigma}_1})\sqrt{\mathbf{\Sigma}_1}\mathbf{V}_1^\top + (\mathbf{U}_2 \sqrt{\mathbf{\Sigma}_2})\sqrt{\mathbf{\Sigma}_2}\mathbf{V}_2^\top \\
&= \left[\mathbf{U}_1 \sqrt{\mathbf{\Sigma}_1}, \mathbf{U}_2 \sqrt{\mathbf{\Sigma}_2}\right]^{m \times (r+r)} \begin{bmatrix} \sqrt{\mathbf{\Sigma}_1}\mathbf{V}_1^\top \\ \sqrt{\mathbf{\Sigma}_2}\mathbf{V}_2^\top \end{bmatrix}^{(r+r) \times n} \\
&\stackrel{\text{SVD}}{=\!=\!=} (\mathbf{U}_3 \mathbf{\Sigma}_3 \mathbf{V}_3^\top)(\mathbf{U}_4 \mathbf{\Sigma}_4 \mathbf{V}_4^\top) \\
&= (\mathbf{U}_3 \mathbf{\Sigma}_3 \mathbf{V}_3^\top \mathbf{U}_4 \sqrt{\mathbf{\Sigma}_4})_{:,:r}(\sqrt{\mathbf{\Sigma}_4}\mathbf{V}_4^\top)_{:r,:} \\
&= \hat{\mathbf{B}}\hat{\mathbf{A}}
\end{aligned} \tag{15}
$$

### D.5 PROOF OF THE LOSSLESSNESS OF MEANINGFUL MODEL UPDATES IN SHE-LoRA

The losslessness of meaningful model updates in the aggregation of SHE-LoRA is supported by the following theorem.

**Theorem 1** *Whether a column of the parameter matrix is encrypted or not, it will always be integrated into the aggregated model, and hence results in no loss of meaningful model updates.*

**Proof 1** *Suppose that for clients 1 to $N$, their encryption budgets are $\gamma_i$ ($\gamma_1 \leq \gamma_2 \cdots \leq \gamma_N$), and the hidden size of the model is $n$ (i.e., number of columns in the parameter matrix). Then the numbers of encrypted columns of the clients are $k_1 = n \times \gamma_1 \leq k_2 = n \times \gamma_2 \cdots \leq k_N = n \times \gamma_N$.*

*From Section 3.3, all $k_i$ encrypted columns are integrated in $\Delta\overline{\mathbf{W}}_{cipher}$, while plaintext columns are integrated in $\Delta\overline{\mathbf{W}}_{plain}$. According to Section 3.4, $\Delta\overline{\mathbf{W}}_{plain} \overset{SVD}{=} \mathbf{B}_p\mathbf{A}_p$ and $\Delta\overline{\mathbf{W}}_{cipher} \overset{SVD}{=} \mathbf{B}_c\mathbf{A}_c$, respectively. Following Eq. (4), all meaningful updates are integrated in LoRA matrices as $\mathbf{B} = [\mathbf{B}_p \quad \mathbf{B}_c]$ and $\mathbf{A} = [\mathbf{A}_p \quad \mathbf{A}_c]^\top$. Finally, the weight update for each client can be calculated as $\Delta\overline{\mathbf{W}} = \Delta\overline{\mathbf{W}}_{plain} + \Delta\overline{\mathbf{W}}_{cipher} = [\mathbf{B}_p \quad \mathbf{B}_c][\mathbf{A}_p \quad \mathbf{A}_c]^\top = \mathbf{B}\mathbf{A}$. Thus, whether a column is encrypted (in $\Delta\overline{\mathbf{W}}_{cipher}$) or not (in $\Delta\overline{\mathbf{W}}_{plain}$), it will always be integrated into the aggregated model ($\Delta\overline{\mathbf{W}} = \mathbf{B}\mathbf{A}$).*

### D.6 DISTRIBUTION SHIFT OF MODEL PARAMETER IMPORTANCE VALUES

To determine whether the distribution of model parameter importance values will shift during training, we conduct 50 rounds of FL training on the Natural-Instructions (Wang et al., 2022) dataset under Non-IID conditions with the Dirichlet distribution parameter $\rho = 0.3$. Fig. 11 illustrates the variation of the distribution of channel-wise importance values along with the progress of FL training. We can see that the specific importance values do change slightly, but their relative ranking remains almost unchanged as compared to Fig. 4. Considering that the parameter importance in SHE-LoRA is assessed via channel-wise summation of sensitivity values, the slight change of model parameter importance distribution has minor impact on performance.

Moreover, considering that extreme cases (e.g., dynamic data change) may occur, especially under Non-IID settings, the negotiation of HE subsets can be executed periodically depending on the clients' tolerance to the change of model parameter importance distribution. Specifically, the theoretical costs of negotiation and training per layer on a client are listed in Table 3.

Table 3: Theoretical costs of negotiation and training per layer.

| | Communication | Computation |
|---|---|---|
| Negotiation $\times N^r$ | 4 Bytes $\times$ hidden size $\times$ ratio $\times N^r$ | Forward $\times N^r$ |
| Training $\times 1$ | 2 Bytes $\times$ rank $\times$ hidden size | Backpropagation $\times 1$ |

With precision=bf16, encryption ratio=1% and $r$=16, FL training generally takes $N^r < 50$ rounds for convergence. Even if the negotiation is executed per round, the negotiation communication overhead of 50 rounds is 4 Bytes$\times$6656 hidden size (Llama-30B)$\times$1%$\times$50=13 KB, which is much smaller than the overall training communication overhead (2 Bytes$\times$16$\times$6656 hidden size (Llama-30B)=213 KB). However, although the computation cost of a "Forward" is much lower than that of a "Backpropagation", the computation cost of N rounds of "Forward" will gradually increase along with the training progress. Therefore, the clients can choose the negotiation period according to their expected balance between model parameter importance update timeliness and computation cost.

### D.7 PROOF OF ASYMPTOTIC GAUSSIAN-DISTRIBUTED NOISE

Let $\mathbf{G} = [g_1, \ldots, g_n] \in \mathbb{R}^{r \times n}$ be the gradient matrix from LoRA fine-tuning, with columns $g_k \in \mathbb{R}^r$, and let $\mathbf{P}$ be the permutation matrix corresponding to a uniform random permutation $\pi(\cdot)$. We define the noise matrix as $\mathbf{\Theta} = \mathbf{G}(\mathbf{P} - \mathbf{I})$, so that the noise added on each gradient element is calculated as $\sigma_{i,k} = g_{i,\pi(k)} - g_{i,k}$. For any fixed linear query matrix $\mathbf{Q} \in \mathbb{R}^{r \times n}$, the query output is:

$$O_{\mathbf{Q}}(\mathbf{\Theta}) = \langle \mathbf{Q}, \mathbf{\Theta} \rangle = \sum_{i=1}^{r} \sum_{k=1}^{n} \mathbf{Q}_{i,k}(g_{i,\pi(k)} - g_{i,k}). \tag{16}$$

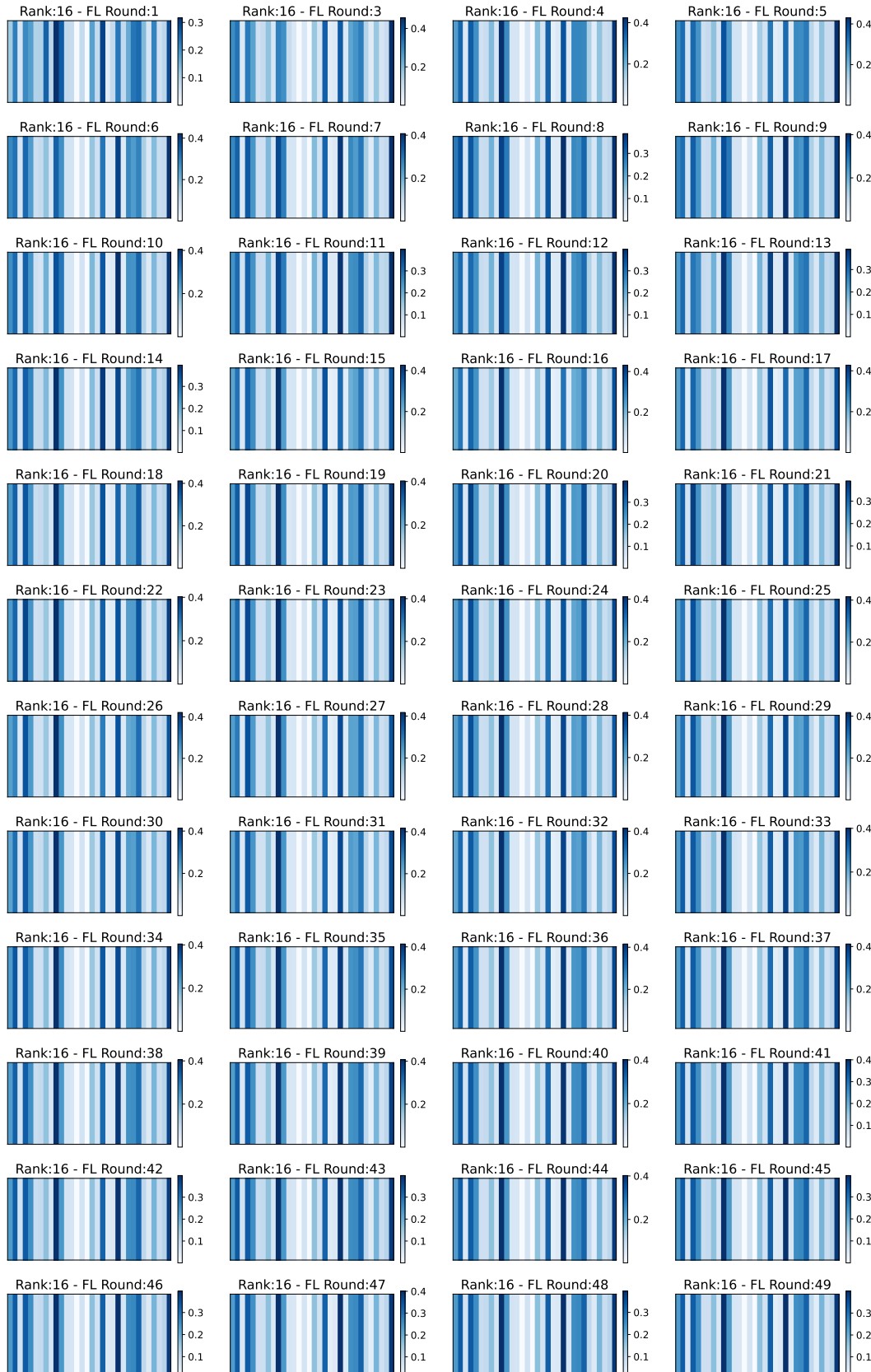

Figure 11: Distribution of model parameter importance values across FL rounds.

The randomness comes solely from the permuted sum $\sum_{i,k} \mathbf{Q}_{i,k} g_{i,\pi(k)}$, which is a combinatorial statistic of the form studied in (Hoeffding, 1951). Therefore, by combinatorial central limited theorem (Hoeffding, 1951) and Berry–Esseen bound (Bolthausen, 1984), we have

$$\frac{O_{\mathbf{Q}}(\boldsymbol{\Theta})}{\sqrt{\mathrm{Var}[O_{\mathbf{Q}}(\boldsymbol{\Theta})]}} \xrightarrow{\mathrm{dist}} \mathcal{N}(0,1), \quad \sup_{x \in \mathbb{R}} \left| \mathcal{P}\left( \frac{O_{\mathbf{Q}}(\boldsymbol{\Theta})}{\sqrt{\mathrm{Var}[O_{\mathbf{Q}}(\boldsymbol{\Theta})]}} \leq x \right) - \Phi(x) \right| = O\left( \frac{1}{\sqrt{n}} \right). \quad (17)$$

The first statement means that the response to any fixed linear query is asymptotically Gaussian, and the second quantifies the approximation error as $O(1/\sqrt{n})$. Moreover, by the classical variance formula for permuted linear statistics (Hoeffding, 1951; Hájek, 1961), we have

$$\mathrm{Var}[O_{\mathbf{Q}}(\boldsymbol{\Theta})] = \sum_{i=1}^{r} \frac{1}{n-1} \left( \sum_{k=1}^{n} (\mathbf{Q}_{i,k} - \bar{\mathbf{Q}}_i)^2 \right) \left( \sum_{k=1}^{n} (g_{i,k} - \bar{g}_i)^2 \right), \quad (18)$$

where $\bar{g}_i = \frac{1}{n} \sum_{k=1}^{n} g_{i,k}$ and $\bar{\mathbf{Q}}_i = \frac{1}{n} \sum_{k=1}^{n} \mathbf{Q}_{i,k}$.

For fixed $\mathbf{G}$ and $\mathbf{Q}$, we define $s^2$ as the variance of $O_{\mathbf{Q}}(\boldsymbol{\Theta})$. Since this asymptotic normality holds for every fixed query matrix $Q$, the Cramér–Wold device (Cramér & Wold, 1936) implies that the noise matrix $\boldsymbol{\Theta}$, viewed as a random vector in $\mathbb{R}^{rn}$, converges in distribution to a zero-mean multivariate Gaussian with the same covariance structure as $\boldsymbol{\Theta}$. Consequently, $\boldsymbol{\Theta}$ behaves like Gaussian noise for all linear queries.

## E    DISTRIBUTABILITY AND SCALABILITY

Our system implementation is predicated upon Flower, an open-source FL framework developed by a team at the University of Oxford. Flower is designed to streamline the construction of FL systems while affording a high degree of flexibility and scalability. It supports a variety of mainstream machine learning frameworks, such as PyTorch, TensorFlow, and Hugging Face Transformers, rendering it suitable for researchers and engineers addressing FL requirements across diverse scenarios. Flower allows users to extensively configure the framework according to their specific needs, thereby accommodating various FL scenarios while offering substantial support for AI research. Based on Flower, our SHE-LoRA supports parallelized simulation and multi-machine deployment, capable of satisfying the distributed and scalable requirements inherent in real-world applications.

### E.1    PERFORMANCE ON MORE CLIENTS

We repeat the experiment of bert-large model with 100, 300, 500, 1000, 2000 clients on the IMDB datasets, which takes 1.58, 4.14, 6.91, 13.8 and 25.3 hours to complete 10 rounds of FL training. Each client encrypts 0.5% of OpenLLaMA-3B with the same rank (16). The means and standard deviations of HE time and communication overhead are listed in Table 4.

Table 4: HE time and communication overhead on varying number of clients.

| # of Clients | HE Time | Communication Overhead |
| --- | --- | --- |
| 100 | 53.34±0.5s | 93.39±0.1MB |
| 300 | 53.57±0.5s | 93.39±0.1MB |
| 500 | 53.49±0.6s | 93.39±0.1MB |
| 1000 | 54.23±0.8s | 93.39±0.1MB |
| 2000 | 54.19±1.1s | 93.39±0.1MB |

We find that although the convergence of FL training does slow down along with the increase of clients, thanks to SHE-LoRA's global control over HE subset, the clients' HE time and communication overhead do not significantly inflate even in extreme heterogeneity with >1000 clients. This means that SHE-LoRA will not delay FL training and scales well with increased number of clients.

## E.2 PERFORMANCE ON LARGER LLMS

We deploy SHE-LoRA on larger LLMs including OpenLlama-3B, Llama-3-8B, Llama-30B and Llama-3.1-70B, and analyze its scalability in comparison with the DP baseline. Specifically, as Section 4.4 has confirmed that SHE-LoRA is secure against the DAGER attack as long as more than 0.125% of the parameters are encrypted, we let each client encrypt 0.125% of the parameters in the scalability experiments with rank $r$=16. In the DP baseline, we let each client add DP noise to parameters with $(\epsilon, \sigma) = (10, 10^{-7})$, which is the same as in the DAGER (Petrov et al., 2024) experiments. The HE key size for OpenLlama-3B and Llama-3-8B is set to 8192. However, the HE key size for Llama-30B and Llama-70B is set to 16384 as the HE key size of 8192 cannot hold a single column (minimum encryption unit in SHE-LoRA) of LLMs at this scale. Then, we measure the encryption time, time cost with DP and ciphertext size per client under varying model scales. The mean and standard deviation of the measured results are shown in Table 5.

Table 5: The costs under varying model scales.

|  | OpenLlama-3B | Llama-3-8B | Llama-30B | Llama-3.1-70B |
| --- | --- | --- | --- | --- |
| # of Layers | 26 | 32 | 60 | 80 |
| Hidden Size | 3200 | 4096 | 6656 | 8192 |
| Encryption Budget | 0.125% | 0.125% | 0.125% | 0.125% |
| Encrypted Parameters | 1,664 | 2,624 | 8,040 | 13,120 |
| Encryption Time (s) | 2.67±0.32 | 4.46±0.73 | 148.70±5.41 | 242.32±8.72 |
| HE Key Size | 8192 | 8192 | 16384 | 16384 |
| Time Cost with DP (s) | 0.0548±0.001 | 0.0878±0.005 | 0.2693±0.028 | 0.4575±0.061 |
| Ciphertext Size (MB) | 23.34±0.00 | 35.91±0.01 | 289.37±0.01 | 385.84±0.03 |

When HE key size is 8192, for OpenLlama-3B, encryption time per parameter=2.67/1664=0.0016 s, ciphertext size per parameter=23.34/1664=0.0140 MB; for Llama-3-8B, encryption time per parameter=4.46/2624=0.0017 s, ciphertext size per parameter=35.91/2624=0.0137 MB.

When HE key size is 16384, for Llama-30B, encryption time per parameter=148.70/8040=0.0185 s, ciphertext size per parameter=289.37/8040=0.0360 MB; for Llama-3.1-70B, encryption time per parameter=242.32/13120=0.0184 s, ciphertext size per parameter=385.84/13120=0.0294 MB.

These observations demonstrate that when LLMs are encrypted with the same level of HE key size, the encryption time and ciphertext size scale almost linearly with the increase of LLM scale. Although the time cost with DP is much lower than that of SHE-LoRA, DP may significantly degrade model accuracy (Sun et al., 2024c), and is vulnerable against inversion attacks under low-noise-level settings (as shown in Table 10 of (Petrov et al., 2024) and Table 2)]).

## E.3 PERFORMANCE ON STRONGER BASE MODELS AND MORE CHALLENGING BENCHMARKS

We conduct the fine-tuning of Qwen3-4B-Instruct-2507 [3] and Llama-3.2-3B [4] with SHE-LoRA and Vanilla LoRA on the PILE dataset [5], and evaluate the performance of the fine-tuned models on six benchmarks: MMLU-Pro [6], GPQA [7], MuSR [8], MATH [9], IFEval [10], and BBH [11]. The collected results of SHE-LoRA, Vanilla LoRA and the base model without fine-tuning are shown in Table 6.

The results demonstrate that SHE-LoRA preserves the original LoRA performance while providing privacy via selective HE. This benefit generalizes across stronger base model families and diverse tasks, provided that federated LoRA is used for PEFT.

---

[3] https://huggingface.co/Qwen/Qwen3-4B-Instruct-2507
[4] https://huggingface.co/meta-llama/Llama-3.2-3B
[5] https://huggingface.co/datasets/iamgroot42/mimir
[6] https://huggingface.co/datasets/TIGER-Lab/MMLU-Pro
[7] https://github.com/idavidrein/gpqa
[8] https://github.com/Zayne-sprague/MuSR
[9] https://huggingface.co/datasets/nlile/hendrycks-MATH-benchmark
[10] https://huggingface.co/datasets/google/IFEval
[11] https://github.com/suzgunmirac/BIG-Bench-Hard

Table 6: Performance comparison of Qwen-3 and Llama-3.2 across more challenging benchmarks.

| Method | Method | MMLU-Pro | GPQA | MuSR | MATH | IFEval | BBH |
|---|---|---|---|---|---|---|---|
| Qwen3-4B | SHE-LoRA | 47.36 | 43.94 | 30.71 | 78.86 | 67.34 | 63.26 |
| | Vanilla LoRA | 48.54 | 42.57 | 32.31 | 77.63 | 68.06 | 64.58 |
| | Base | 65.36 | 45.00 | 61.67 | 84.00 | 90.17 | 85.93 |
| Llama-3.2-3B | SHE-LoRA | 14.86 | 13.64 | 17.86 | 21.74 | 27.24 | 9.26 |
| | Vanilla LoRA | 14.62 | 13.87 | 18.05 | 20.96 | 27.37 | 9.38 |
| | Base | 14.29 | 11.11 | 30.95 | 16.68 | 27.99 | 10.85 |

# F  ADDITIONAL EXPERIMENTAL RESULTS

## F.1  PERFORMANCE ON VARYING TASKS

We employ FedIT (Zhang et al., 2024) and FedSA (Guo et al., 2024) as baseline methods under homogeneous settings (rank $r$=8). FedIT averages LoRA weights across clients, limiting the rank according to the capability of the weakest device. FedSA trains matrix $\mathbf{B}$ locally while aggregating matrix $\mathbf{A}$ globally, leveraging FL to enhance the representation capacity of LoRA. Moreover, we employ FLoRA (Wang et al., 2024), HeterLoRA (Cho et al., 2024) and Flex-LoRA (Bai et al., 2024) as baselines under heterogeneous settings. FLoRA utilizes stacking to reduce full-weight computation and achieve precise averaging across heterogeneous LoRA updates, but at the cost of an expanded parameter space. HeterLoRA zero-pads all LoRA matrices to the global maximum rank, applies weight-averaged aggregation similar to FedAvg, and subsequently truncates the aggregated weights to align with the local rank of each client. However, zero-padding introduces additional dilution in the aggregated parameters, which in turn leads to degraded model performance. Flex-LoRA reconstructs the full parameter matrix for each client by computing $B \times A$ and performs aggregation. Subsequently, the aggregated matrix is decomposed using SVD and truncated according to the client's LoRA rank, producing a low-rank parameter matrix.

### F.1.1  RESULTS ON NLP TASKS

**Natural Language Generation**: According to the results in Table 7, FedIT and FedSA perform the worst on the MMLU Benchmark, obtaining scores of 21.2 and 20.1, respectively. These results indicate the limitations of traditional homogeneous approaches in heterogeneous LoRA settings, where the inability to effectively utilize client-specific information hinders overall performance. While HeterLoRA integrates parameters from heterogeneous devices to improve performance, its reliance on zero-padding leads to parameter dilution, resulting in inferior performance compared to Flex-LoRA. SHE-LoRA achieves the highest scores on STEM, Social Sciences(SS) and the overall Average, and matches Flex-LoRA's performance on Humanities. Both methods outperform all other baselines by a significant margin. These results indicate that SHE-LoRA better preserves informative updates in heterogeneous generative tasks, leading to improved generalization and performance.

Table 7: Performance on the MMLU benchmark.

| Method | STEM | SS | Humanities | Average |
|---|---|---|---|---|
| FedIT (Zhang et al., 2024) | 21.5 | 21.3 | 20.4 | 21.2 |
| FedSA (Guo et al., 2024) | 21.8 | 21.4 | 19.7 | 20.1 |
| HeterLoRA (Cho et al., 2024) | 24.7 | 25.4 | 25.8 | 26 |
| Flex-LoRA (Bai et al., 2024) | 26.2 | 27.9 | **26.6** | 27.4 |
| SHE-LoRA | **28.1** | **29.2** | 26.5 | **28.2** |

**Natural Language Understanding**: Similarly, we reviewed on the six datasets of GLUE Benchmark in Table 8, and the performances of FedIT and FedSA reaffirmed the limitations of traditional aggregation methods in heterogeneous scenarios. Flex-LoRA and SHE-LoRA, on the other hand, outperform the other methods, demonstrating that SHE-LoRA can more effectively update model parameters in heterogeneous environments while achieving performance comparable to non-private methods. Unsurprisingly, HeterLoRA achieves better performance than homogeneous baselines. However, it lags behind Flex-LoRA and SHE-LoRA, primarily due to the performance degradation caused by parameter dilution.

Table 8: Performance on the GLUE benchmark.

| Method | SST2 | MRPC | QQP | RTE | WNLI | QNLI |
|---|---|---|---|---|---|---|
| FedIT (Zhang et al., 2024) | 47.41 | 31.62 | 64.71 | 43.07 | 46.34 | 48.87 |
| FedSA (Guo et al., 2024) | 48.23 | 33.71 | 66.32 | 43.56 | 48.27 | 48.26 |
| HeterLoRA (Cho et al., 2024) | 55.73 | 68.38 | 72.17 | 44.72 | 48.86 | 49.14 |
| Flex-LoRA (Bai et al., 2024) | 52.29 | **74.81** | **75.31** | 46.93 | 49.66 | 49.51 |
| SHE-LoRA | **57.11** | 70.88 | 72.52 | **50.18** | **57.75** | **59.63** |

SHE-LoRA demonstrates strong performance across both benchmarks, achieving SOTA results in heterogeneous settings while maintaining optimal performance despite the integration of privacy-preserving mechanisms.

Table 9: Performance comparison on 5 vision tasks.

| Method | Datasets | | | | | |
|---|---|---|---|---|---|---|
| | MNIST | DTD | EuroSAT | GTSRB | SVHN | AVG |
| *Clip-Vit-Base-Patch-16* r = 8 | | | | | | |
| FedIT (Zhang et al., 2024) | 93.38 | 68.74 | 93.17 | 83.62 | 90.43 | 85.87 |
| FedSA (Guo et al., 2024) | 93.13 | 67.51 | 94.23 | 85.12 | 88.49 | 85.69 |
| HeterLoRA (Cho et al., 2024) | 95.37 | 68.83 | 96.22 | 87.18 | 91.55 | 87.83 |
| Flex-LoRA (Bai et al., 2024) | 99.28 | **70.32** | **98.48** | 95.74 | **95.37** | **91.84** |
| SHE-LoRA | **99.33** | 69.97 | 98.35 | **95.88** | 95.13 | 91.73 |
| *Clip-Vit-Base-Patch-16* r = 16 | | | | | | |
| FedIT (Zhang et al., 2024) | 95.36 | 68.85 | 94.56 | 85.37 | 91.58 | 87.14 |
| FedSA (Guo et al., 2024) | 94.62 | 67.92 | 95.18 | 87.23 | 90.67 | 87.12 |
| HeterLoRA (Cho et al., 2024) | 94.56 | 68.21 | 96.77 | 89.62 | 92.28 | 88.29 |
| Flex-LoRA (Bai et al., 2024) | **99.30** | 70.05 | **98.29** | **95.45** | 95.15 | 91.65 |
| SHE-LoRA | 99.25 | **70.85** | 98.22 | 95.35 | **96.03** | **91.94** |

### F.1.2 RESULTS ON VISION TASKS

We apply CLIP (Radford et al., 2021) as the basic pre-trained model for vision tasks, a multimodal model that mixes visual model and language model. Specifically, we load the Clip-Vit-Base-Patch-16 model from huggingface[12] and fine-tune its visual model, and conduct experiments on five visual classification tasks, which are MNIST (LeCun et al., 2002), DTD (Cimpoi et al., 2014), EuroSAT (Helber et al., 2019), GTSRB (Stallkamp et al., 2012), SVHN (Netzer et al., 2011). We conduct FL training for 10 rounds on each task, and set that each client has the same LoRA rank.

The results are shown in Table 9. The highest accuracy (%) for each task is highlighted in **blod**. At rank $r = 8$, SHE-LoRA achieves a comparable average accuracy (91.73%) to that of Flex-LoRA (91.84%), while outperforming FedIT, FedSA and HeterLoRA. At the rank of $r = 16$, SHE-LoRA can even achieve the best average accuracy (91.94%). The results indicate that the privacy protection mechanism of SHE-LoRA will not lead to significant performance degradation.

### F.2 ROBUSTNESS UNDER VARYING NON-IID CONDITIONS

The results of SHE-LoRA in Tables 7 and 8 are collected under the Dirichlet distribution with parameter $\rho$=0.3, which confirm that SHE-LoRA achieves comparable performance to a SOTA non-private Federated PEFT method (Flex-LoRA) on various benchmarks under Non-IID conditions.

To further validate the robustness of SHE-LoRA under varying Non-IID conditions, we conduct more experiments on the natural-instructions dataset with $\rho$ set to 0.1, 0.5, 1 and 10, respectively. A smaller $\rho$ indicates a greater Non-IID degree among clients. The experiment is repeated for 10 rounds under each $\rho$ value. The mean, standard deviation of model accuracies collected on the MMLU Benchmark are shown in Table 10 ($\uparrow$ means that higher accuracy is better):

---

[12]https://huggingface.co/openai/clip-vit-base-patch16

Table 10: MMLU benchmark under varying Non-IID conditions.

| $\rho$ | STEM↑ | SS↑ | Humanities↑ | Average↑ |
|---|---|---|---|---|
| 0.1 | 24.8±0.00 | 25.5±0.35 | 25.4±0.15 | 25.9±0.09 |
| 0.5 | 24.7±0.15 | 25.6±0.46 | 25.4±0.21 | 25.8±0.21 |
| 1 | 24.7±0.51 | 25.4±0.11 | 25.3±0.25 | 25.8±0.06 |
| 10 | 24.8±0.21 | 25.4±0.12 | 25.5±0.11 | 25.8±0.10 |

We can see that no matter how Non-IID the clients' data is, the models trained with SHE-LoRA can achieve stable performance across clients, which validates the robustness of SHE-LoRA under varying Non-IID conditions.

### F.3 EFFICIENT ESTIMATION OF MUTUAL INFORMATION

As described in Section 2.2, mutual information measures the amount of information shared between two variables. According to Eq. (3), evaluating the mutual information requires knowledge of $p(x)$, $p(y)$ and the joint density $p(x, y)$, yet in practice we have only samples and not the true densities. The simplest empirical approach is a histogram (binning) estimator, which partitions the space and counts frequencies. However, histograms require large sample sizes and are sensitive to the choice of binning. A more stable nonparametric approach is to employ kernel density estimators (KDE) (Moon et al., 1995). Concretely, flatten the parameter matrices $\mathbf{W}$ and $\mathbf{W}_{-\mathbf{w}}$ into one-dimensional collections and treat the corresponding elements as paired samples $\{(x_i, y_i)\}_{i=1}^N$. In this step, the marginal distribution of $p(x)$, $p(y)$ and the joint density $p(x, y)$ are estimated by kernel density estimation (KDE), which constructs a smooth probability density function by centering kernel functions (e.g., Gaussian) at each sample point and aggregating them with an appropriate bandwidth. Once these probability density variables are obtained, they are substituted into Eq. (3) to compute the final mutual information. The code for the mutual information calculation is given as follows:

```python
from sklearn.neighbors import KernelDensity
def kde_mutual_info(X_flat, Y_flat, bandwidth=0.2):
    X_flat = X_flat.flatten()
    Y_flat = Y_flat.flatten()
    n = len(X_flat)
    sample_num = min(10000, n)
    sample_points = np.random.choice(n, sample_num, replace=False)
    X_sample = X_flat[sample_points].reshape(-1, 1)
    Y_sample = Y_flat[sample_points].reshape(-1, 1)
    XY_sample = np.hstack([X_sample, Y_sample])
    kde_x = KernelDensity(bandwidth=bandwidth).fit(X_sample)
    kde_y = KernelDensity(bandwidth=bandwidth).fit(Y_sample)
    kde_xy = KernelDensity(bandwidth=bandwidth).fit(XY_sample)
    log_px = kde_x.score_samples(X_sample)
    log_py = kde_y.score_samples(Y_sample)
    log_pxy = kde_xy.score_samples(XY_sample)
    return np.mean(log_pxy - log_px - log_py)
```

### F.4 RESISTANCE AGAINST MEMBERSHIP INFERENCE ATTACKS

We fine-tune the base model Qwen3-4B-Instruct-2507 on the PILE dataset using standard LoRA (denoted as *Vanilla LoRA* in tables) and SHE-LoRA with varying encryption ratios $\gamma$, respectively. Then, we implement seven membership inference attacks (MIAs)(Loss (Carlini et al., 2021), Lower-case (Carlini et al., 2021), Zlib (Carlini et al., 2021), Min-k (0.1) (Shi et al., 2024), Min-k (0.5) (Shi et al., 2024), Recall (Xie et al., 2024) and PAC (Ye et al., 2024)) on the base model (denoted as *Base* in tables) and the fine-tuned models of Vanilla LoRA and SHE-LoRA. Under SHE-LoRA, attackers can only launch MIAs based on unencrypted parameters. The attack results are reported in AUROC with Table 11, FPR@95 with Table 12 and TPR@5 with Table 13.

In the AUROC results, *Base* performs no better than random guessing (with AUROC≈50%), confirming that the pretraining corpus does not include the evaluation data. In contrast, Vanilla LoRA achieves much higher AUROC results across all attacks, indicating substantial membership leakage after fine-tuning. Remarkably, compared with Vanilla LoRA, SHE-LoRA reduces the average MIA

Table 11: The AUROC results reported under 7 membership inference attacks.

| Model | Loss | Lowercase | Zlib | Min-k (0.1) | Min-k (0.5) | Recall | PAC |
|---|---|---|---|---|---|---|---|
| Base | 50.9% | 48.4% | 50.2% | 50.5% | 50.9% | 50.1% | 51.2% |
| Vanilla LoRA | 81.4% | 80.5% | 76.7% | 80.9% | 82.9% | 73.8% | 83.3% |
| $\gamma = 1‰$ | 62.6% | 62.8% | 60.3% | 62.5% | 63.5% | 64.7% | 65.0% |
| $\gamma = 1\%$ | 56.8% | 57.7% | 55.4% | 56.5% | 57.3% | 58.4% | 58.4% |
| $\gamma = 5\%$ | 54.1% | 55.2% | 53.1% | 53.7% | 54.3% | 55.8% | 55.3% |
| $\gamma = 10\%$ | 56.8% | 57.7% | 55.4% | 56.5% | 57.3% | 58.4% | 58.4% |
| $\gamma = 20\%$ | 52.4% | 53.2% | 51.7% | 52.1% | 52.5% | 53.1% | 53.3% |

success rate by 21.0% with an encryption ratio as low as $\gamma = 1‰$. With the increasing of $\gamma$ (e.g., to 1%), attack success rates further drop by 20.9% 30.9%, resulting in nearly random-guessing performance and demonstrating significantly stronger privacy protection.

Table 12: The FPR@95 results reported under 7 membership inference attacks.

| Model | Loss | Lowercase | Zlib | Min-k (0.1) | Min-k (0.5) | Recall | PAC |
|---|---|---|---|---|---|---|---|
| Base | 94.1% | 95.4% | 96.1% | 96.1% | 95.2% | 95.5% | 95.7% |
| Vanilla LoRA | 66.5% | 63.3% | 88.8% | 71.4% | 66.2% | 79.2% | 72.1% |
| $\gamma = 1‰$ | 89.7% | 86.9% | 93.9% | 90.8% | 90.3% | 90.5% | 89.9% |
| $\gamma = 1\%$ | 92.4% | 90.7% | 95.2% | 93.8% | 93.4% | 91.6% | 93.1% |
| $\gamma = 5\%$ | 93.2% | 92.1% | 95.5% | 94.5% | 94.1% | 93.1% | 93.8% |
| $\gamma = 10\%$ | 92.4% | 90.7% | 95.2% | 93.8% | 93.4% | 91.6% | 93.1% |
| $\gamma = 20\%$ | 93.2% | 93.5% | 95.6% | 95.4% | 95.1% | 94.0% | 94.8% |

Table 13: The TPR@5 results reported under 7 membership inference attacks.

| Model | Loss | Lowercase | Zlib | Min-k (0.1) | Min-k (0.5) | Recall | PAC |
|---|---|---|---|---|---|---|---|
| Base | 8.6% | 5.6% | 7.4% | 5.5% | 8.0% | 5.1% | 9.9% |
| Vanilla LoRA | 35.7% | 35.6% | 39.7% | 39.4% | 41.8% | 25.4% | 53.5% |
| $\gamma = 1‰$ | 12.5% | 13.0% | 15.2% | 14.0% | 14.5% | 15.4% | 17.4% |
| $\gamma = 1\%$ | 9.7% | 9.3% | 10.8% | 9.1% | 10.2% | 10.3% | 15.0% |
| $\gamma = 5\%$ | 9.0% | 7.4% | 9.1% | 7.3% | 9.3% | 7.7% | 12.4% |
| $\gamma = 10\%$ | 9.7% | 9.3% | 10.8% | 9.1% | 10.2% | 10.3% | 11.9% |
| $\gamma = 20\%$ | 8.2% | 6.7% | 8.5% | 6.5% | 8.2% | 6.2% | 11.1% |

Consistent with the AUROC results, even at $\gamma = 1‰$, SHE-LoRA achieves an average FPR@95 of 90.27% and an average TPR@5 of 14.57%, closely comparable to the base model's performance (FPR@95=95.44%, TPR@5=7.16%). In contrast, Vanilla LoRA is significantly more vulnerable, with an average FPR@95 of 72.50% and TPR@5 of 38.73%. These results demonstrate that SHE-LoRA preserves membership privacy during fine-tuning: even under a very small encryption ratio, attackers can only achieve performance close to random guessing, with negligible advantage in distinguishing members from non-members.

In summary, experiments on Qwen3-4B-Instruct-2507 across seven MIAs demonstrate that SHE-LoRA is consistently robust. This stems from two key design features: 1) selective encryption of the most sensitive parameter columns prevents direct leakage of privacy-critical information, and 2) column-wise position obfuscation, similar to injecting structured perturbations (see Appendix D.7), increases uncertainty for attackers. These mechanisms also mitigate property inference and reconstruction attacks that leverage auxiliary priors, as they obscure the very gradients or parameters these attacks typically exploit.

## F.5 IMPACT OF SENSITIVE PARAMETERS ON PERFORMANCE

As theoretically established in Appendix B.1, parameter sensitivity is closely linked to privacy risk. To empirically validate this connection, we conduct experiments on the PILE dataset using the Qwen3-4B-Instruct-2507 model. Specifically, we fine-tune the model on the training set and

Table 14: Model performance comparison with perplexity.

| Model | Base | Vanilla LoRA | $\gamma = 1\text{‰}$ | $\gamma = 1\%$ | $\gamma = 5\%$ | $\gamma = 10\%$ |
|-------|------|--------------|------|------|------|------|
| PPL | 74.01 | 21.23 | 38.83 / 21.23 | 50.53 / 21.23 | 55.21 / 21.23 | 60.57 / 21.23 |

evaluate text generation quality via perplexity (PPL) on the validation set. A higher PPL indicates poorer adaptation to the target domain. We compare three settings Table 14: (i) the original base model (denoted as *Base*), (ii) a raw LoRA-finetuned model (denoted as *Vanilla LoRA*), and (iii) SHE-LoRA with varying encryption ratios $\gamma$. This allows us to assess whether protecting high-sensitivity parameters, rather than removing or ignoring them, preserves model utility while enhancing privacy.

SHE-LoRA reports two PPL metrics: the value on the left of "/" reflects the model performance when encrypted parameters are masked (i.e., using only unencrypted columns), while the value on the right of "/" reflects the model's true performance without masking parameters. As expected, *Base* exhibits high PPL due to lack of domain adaptation, whereas *Vanilla LoRA* significantly reduces PPL, confirming effective learning. Notably, SHE-LoRA with an encryption ratio of merely 1‰ already raises the masked PPL to 38.83, indicating that even trivial removal of the most sensitive columns substantially degrades utility. In contrast, the PPL result on the right of "/" is nearly identical to that of *Vanilla LoRA*, demonstrating sound utility preservation under SHE-LoRA. Furthermore, as the encryption ratio increases, masked PPL consistently rises, confirming that SHE-LoRA prioritizes the most privacy-sensitive columns.

## G  TABLE OF NOTATIONS

Table 15 lists the main notations used in this paper.

Table 15: Table of Notations

| Notation | Description |
|----------|-------------|
| $\mathbf{W} \in \mathbb{R}^{m \times n}$ | Model parameters of a LLM |
| $\mathbf{W}_{-w} \in \mathbb{R}^{m \times n}$ | Model parameters with $w$ zeroed-out |
| $\mathbf{W}_0 \in \mathbb{R}^{m \times n}$ | Frozen pre-trained parameters |
| $\mathbf{A} \in \mathbb{R}^{r \times n}$ | Low-rank adapter matrix A of LoRA |
| $\mathbf{B} \in \mathbb{R}^{m \times r}$ | Low-rank adapter matrix B of LoRA |
| $\mathbf{X} \in \mathbb{R}^{L \times n}$ | Input embedding |
| $\mathbf{G}$ | Gradient Matrix |
| $\mathcal{L}(\cdot)$ | Loss function |
| $\Omega(\cdot)$ | Sensitivity computation function |
| $\mathcal{S}(\cdot)$ | Selective HE method |
| $\mathcal{R}(\cdot)$ | Any data reconstruction attack method |
| $r$ | Rank of LoRA adapter |
| $L$ | Number of tokens in an input sequence |
| $x_i \in \mathbb{R}^L$ | The $i$-th features in the input |
| $I(W; W_{-w})$ | Mutual information between $W$ and $W_{-w}$ |
| $\gamma_i$ | Ratio of parameters in client $i$ for encryption |
| $k_i$ | Number of columns in client $i$ for encryption |
| $G_i$ | Group of indices of selected columns in client $i$ |
| $S_i$ | Sensitivities of the columns in $G_i$ on client $i$ |
| $b_i$ | Block $i$ of tensor to be encrypted |
| $N^b$ | Number of tensor blocks to be encrypted |
| $pk$ | Public HE key |
| $C_i$ | Ciphertext of the $i$-th block |
| $K$ | Max columns of unencrypted parameters among clients |

## H  LIMITATIONS

As described in Section 2.3 and Appendix C, SHE-LoRA operates under the assumption of an honest-but-curious server, where all clients share the same HE key. Although secure communication channels can be used to defend against malicious clients or collusion between the server and clients, such mechanisms incur higher encryption costs. A promising direction for future work is to explore

more efficient distributed parameter protection using techniques such as threshold homomorphic encryption, multi-key homomorphic encryption, or proxy re-encryption.

## I   BROADER IMPACT

In this work, we leverage parameter sensitivity and SHE to ensure the secure aggregation of federated LoRA against inversion attacks such as DAGER. Such attacks are able to recover the original client data from clients' updates uploaded during federated PEFT, exacerbating privacy concerns and hindering the possibility of FL to extract value from distributed data. Our work offers adaptive and sufficient privacy preservation, while minimizing HE overhead per client in cross-device federated PEFT with LoRA.

Importantly, we find that with more sensitive model parameters being encrypted, the mutual information that can be leaked from the model updates drops dramatically, indicating that it is possible to effectively reduce the risk of privacy leakage in terms of privacy information as long as the sensitive model parameters are correctly encrypted. Our work implies that critical information within the model parameters can be soundly protected against the SOTA attacks by merely encrypting less than 1% of the model parameters. Furthermore, we take into account the heterogeneity of the parameter sensitivity and encryption capabilities across clients, and broadly adapt the cost-effective SHE-LoRA to accommodate clients with diverse data distributions and device capabilities. With these observations, we highlight the feasibility and effectiveness of applying tailored and secure privacy protection for cross-device federated PEFT at much lower overhead compared to existing off-the-shelf privacy protection techniques.

## J   THE USE OF LARGE LANGUAGE MODELS (LLMs)

According to the policies on large language model usage at ICLR 2026, we state that LLMs are only used to help with paper writing, including spell checking, grammar checking, and polish writing.

