# OpenReview forum: "SHE-LoRA: Selective Homomorphic Encryption for Federated Tuning with Heterogeneous LoRA"
_ICLR.cc/2026/Conference — ICLR 2026 Poster_

### Official Review · Reviewer_sB8q · 2025-10-26

**Soundness:** 3
**Presentation:** 2
**Contribution:** 2
**Rating:** 4
**Confidence:** 3

**Summary:**

This paper introduces a new framework for privacy-preserving federated fine-tuning of large language models (LLMs) that balances efficiency, privacy, and heterogeneity among client devices. The proposed SHE-LoRA integrates SHE with LoRA to allow clients with diverse computational capacities and data distributions to securely participate in federated fine-tuning. The method adaptively encrypts only the "most sensitive" parameters based on a parameter sensitivity analysis, using a global negotiation mechanism to determine a subset of the parameters to be encrypted. Besides, the adaptive aggregation (on server side) and reparametrization (on client side) techniques are proposed to enable the FL cycle.

**Strengths:**

1) The paper proposes a framework that provides a strong privacy protection guarantee for the models.
2) The proposed algorithm can save computation cost significantly compared to the existing methods.
3) The proposed algorithm provides some insight about how to define and select important parameters.

**Weaknesses:**

1) The parameter subset selection and negotiation may remain unclear for readers. For example,
* Why is $S_j$  defined in the paper a reasonable metric to evaluate the importance?
* What does "features aggregated across L tokens" mean in the context of the paper?
* If the importance of the parameters changes in the training process, does the selection dynamically adapt to such changes?
2) It is unclear what privacy protection strength this algorithm can provide in the worst case.
* Do the most sensitive parameters equivalent to those that should be protected for privacy protection?
* It seems there is no guarantee that the most sensitive parameters of a client will remain after negotiation. Does it mean the privacy of such a client, if it exists, is more vulnerable than that of others?
* The attack experiments only consider reconstruction attack, but no membership inference attack (which is more closely related to the privacy definition people commonly believe).
3) Given that the encrypted parameters are only a very small fraction, it is unclear whether directly removing those parameters from training can also produce similar model performance. If so, it may become arguable that the
* Some ablation studies about the parameter may help.
4) The writing of the paper can be further improved.
* Some notations and definitions are missing in the paper. For example, the notation of "batch" uses the same notation as the matrix $B$ in LoRA.
* Too many details are omitted in the main text, which makes readers hard to follow.

**Questions:**

Please refer to the question in the Weaknesses section.

---

> ### Author Response · Authors · 2025-11-23
> **Response (1/4)**
>
> Thank you for the careful reading and valuable feedback.
>
> >### **Weakness 1:** The parameter subset selection and negotiation may remain unclear for readers. For example,
> > - Why is $S_j$ defined in the paper a reasonable metric to evaluate the importance?
> > - What does "features aggregated across L tokens" mean in the context of the paper?
> > - If the importance of the parameters changes in the training process, does the selection dynamically adapt to such changes?
>
> **Response 1:**
> - According to Section 2.4 and Figure 1, if we assess parameter-wise sensitivity at the element level, encrypting even a single sensitive parameter necessitates encryption of the entire column containing that parameter during the aggregation of $BA$, leading to ciphertext expansion across the full column.
> In contrast, column-wise sensitivity assessment and selective encryption of important parameter columns avoid ciphertext expansion caused by redundant encryption of sparsely distributed individual sensitive parameters, reduce HE overhead, and improve ciphertext utilization.
> Moreover, as shown in Figure 4, the sensitivity values exhibit conspicuous column-wise distribution patterns, suggesting a strong correlation with specific data channels (columns).
> Therefore, $S_j$ is a reasonable metric to evaluate importance.
> - By saying "features aggregated across L tokens", we mean the $l_2$ norm is calculated over the $L$ elements in $x_j \in \mathbb{R}^{L}$.
> For clarification, we have modified this phrase to "... where $|| x_j|| _2$ is the $l_2$ norm of the $j$-th feature $x_j \in \mathbb{R}^{L}$" in `Section 3.1.1` of the revised manuscript.
> - As shown in Figure 11, the distribution of column-wise parameter sensitivity values remains relatively stable during FL training, as compared to Figure 4.
> Therefore, SHE-LoRA does not trigger renegotiation automatically, but lets clients decide the renegotiation plan according to their tolerance to sensitivity distribution change.
> Once triggered, the theoretical costs of negotiation and training per layer on a client are listed as below, which have also been included in `Appendix D.6`.
>
>     ||Communication|Computation|
>     |-|-|-|
>     |Negotiation$\times$N|4 Bytes$\times$hidden size$\times$ratio$\times$N|Forward$\times$N|
>     |Training$\times$1|2 Bytes$\times$rank$\times$hidden size|Backpropagation$\times$1|
>
>     With bf16 precision, encryption ratio=1%, rank=16 and hidden size for `Llama-30B`=6656, FL training generally takes <50 rounds for convergence.
>     Even if the negotiation is executed per round, the N-round negotiation communication overhead=4 Bytes$\times$6656$\times$1%$\times$50=13 KB << the training communication overhead=2 Bytes$\times$16$\times$6656=213 KB.
>     However, although the computation cost of a `forward` << the cost of a `backpropagation`, *Forward$\times$N* will gradually increase along with the training progress.
>     Therefore, **the clients can choose the negotiation period according to their expected balance between sensitivity update timeliness and computation cost**.

---

> ### Author Response · Authors · 2025-11-23
> **Response (2/4)**
>
> >### **Weakness 2:** It is unclear what privacy protection strength this algorithm can provide in the worst case.
> > - Do the most sensitive parameters equivalent to those that should be protected for privacy protection?
> > - It seems there is no guarantee that the most sensitive parameters of a client will remain after negotiation. Does it mean the privacy of such a client, if it exists, is more vulnerable than that of others?
> > - The attack experiments only consider reconstruction attack, but no membership inference attack (which is more closely related to the privacy definition people commonly believe).
>
> **Response 2:**
> - Indeed, the most sensitive parameters are precisely those that warrant stronger privacy protection.
> Training loss measures how well the model captures the true data distribution, with lower values indicating outputs closer to real data.
> Lots of model pruning methods (including Wanda) leverage this observation, as detailed in `Appendix B.1`, where we further elaborate on the connection between parameter sensitivity and privacy leakage risk.
> Highly sensitive parameters always carry richer information about the training data, making them more vulnerable to privacy attacks. Therefore, they should be prioritized for protection.
>
>     Moreover, as elaborated in **Response 1 to Reviewer 9N3M** and **Response 3 to Reviewer i41f**, we provide a `theoretical lower bound` on the attacker’s data reconstruction error, proving that protecting the most sensitive parameters directly increases this lower bound—thereby enhancing privacy guarantees.
>
> - Under limited HE budgets, resource-constrained clients must reach a balance that extends HE-based privacy protection to as many clients as possible, while minimizing the privacy leakage from each client’s unselected parameters.
> Thus, we formulate this as a bi-objective optimization problem and provide its detailed solution in `Appendix D.1`.
>
>     Specifically, if a client’s top-sensitive column is not selected, **this does not mean the client loses all protection**.
>     The negotiated result will still guide it to encrypt a subset of parameters, achieving an equilibrium between client coverage and privacy leakage across all clients.
>     At this equilibrium, the client can still maintain privacy guarantees, which can be quantified as the lower bound of reconstruction error (as detailed in **Response 1 to Reviewer 9N3M**).
>     Moreover, our experiments show that **encrypting merely $\gamma _i=1‰$ of sensitive parameters** substantially reduces the success rates of diverse inference attacks, enabling SHE-LoRA to provide strong privacy protection even under highly constrained encryption budgets.
>
> - Indeed, membership inference attacks (MIAs) represent a critical class of privacy threats.
> Accordingly, we compare the effectiveness of SHE-LoRA with standard LoRA (denoted as `Vanilla LoRA` in the table) and the base model `Qwen3-4B-Instruct-2507` (denoted as `Base` in the table) in defending against seven MIA methods, and collect the AUROC results as follows (more results are in **Response 1 to Reviewer pYmU** and `Appendix F.4`):
>
>     |Model|Loss[c1]|Lowercase[c1]|Zlib[c1]|Min-k(0.1)[c2]|Min-k(0.5)[c2]|Recall[c3]|PAC[c4]|
>     |-|-|-|-|-|-|-|-|
>     |Base|50.9%|48.4%|50.2%|50.5%|50.9%|50.1%|51.2%|
>     |Vanilla LoRA|81.4%|80.5%|76.7%|80.9%|82.9%|73.8%|83.3%|
>     |$\gamma=1‰$|62.6%|62.8%|60.3%|62.5%|63.5%|64.7%|65.0%|
>     |$\gamma=$ 1%|56.8%|57.7%|55.4%|56.5%|57.3%|58.4%|58.4%|
>     |$\gamma=$ 5%|54.1%|55.2%|53.1%|53.7%|54.3%|55.8%|55.3%|
>     |$\gamma=$ 10%|56.8%|57.7%|55.4%|56.5%|57.3%|58.4%|58.4%|
>     |$\gamma=$ 20%|52.4%|53.2%|51.7%|52.1%|52.5%|53.1%|53.3%|
>
>     In the AUROC results, `Base` performs no better than random guessing (with AUROC$\approx$50%), confirming that the pretraining corpus does not include the evaluation data.
>     In contrast, `Vanilla LoRA` achieves much higher AUROC results across all attacks, indicating substantial membership leakage after fine-tuning.
>     Remarkably, compared with `Vanilla LoRA`, SHE-LoRA reduces the average MIA success rate by 21.0% with an encryption ratio as low as $\gamma = 1‰$.
>     With the increasing of $\gamma$, attack success rates further drop by 20.9%~30.9%, resulting in nearly random-guessing performance and demonstrating significantly stronger privacy protection.
>
>     In summary, **experiments on `Qwen3-4B-Instruct-2507` across seven MIAs demonstrate that SHE-LoRA is consistently robust**.
>     This stems from two key design features: (1) selective encryption of the most sensitive parameter columns prevents direct leakage of privacy-critical information, and (2) column-wise position obfuscation, similar to injecting structured perturbations (see **Response 1 to Reviewer 9N3M**), increases uncertainty for attackers.
>     These mechanisms also mitigate property inference and reconstruction attacks that leverage auxiliary priors, as they obscure the very gradients or parameters these attacks typically exploit.

---

> ### Author Response · Authors · 2025-11-23
> **Response (3/4)**
>
> [c1] [Extracting Training Data from Large Language Models](https://www.usenix.org/system/files/sec21-carlini-extracting.pdf), USENIX Security Symposium 2021.
>
> [c2] [Detecting Pretraining Data from Large Language Models](https://openreview.net/forum?id=zWqr3MQuNs), ICLR 2024.
>
> [c3] [ReCaLL: Membership Inference via Relative Conditional Log-Likelihoods](https://aclanthology.org/2024.emnlp-main.493/), EMNLP 2024.
>
> [c4] [Data Contamination Calibration for Black-box LLMs](https://aclanthology.org/2024.findings-acl.644/), ACL 2024.

---

> ### Author Response · Authors · 2025-11-23
> **Response (4/4)**
>
> >### **Weakness 3:** Given that the encrypted parameters are only a very small fraction, it is unclear whether directly removing those parameters from training can also produce similar model performance. If so, it may become arguable that the
> > - Some ablation studies about the parameter may help.
>
> **Response 3:** As elaborated in **Response 2**, SHE-LoRA prioritizes the protection of the most sensitive parameter columns, which have the greatest impact on model performance.
> So, they cannot be removed or omitted.
> Instead, **they must be secured via HE to simultaneously preserve their computational utility (via `Reparameterization`) and ensure strong privacy guarantees**.
>
> As theoretically established in `Appendix B.1`, parameter sensitivity is closely linked to privacy risk.
> To empirically validate this connection, we conducted experiments on the PILE dataset using the `Qwen3-4B-Instruct-2507` model.
> Specifically, we fine-tuned the model on the training set and evaluated text generation quality via perplexity (PPL) on the validation set.
> A higher PPL indicates poorer adaptation to the target domain.
> We compared three settings in the table below: (i) the original base model (`Base`), (ii) a raw LoRA-finetuned model (`Vanilla LoRA`), and (iii) SHE-LoRA with varying encryption ratios $\gamma$.
> This allows us to assess whether protecting high-sensitivity parameters, rather than removing or ignoring them, preserves model utility while enhancing privacy.
>
> |Model|Base|Vanilla LoRA|$\gamma=$1‰|$\gamma=$1%|$\gamma=$5%|$\gamma=$10%|$\gamma=$20%|
> |-|-|-|-|-|-|-|-|
> |PPL|74.01|21.23|38.83/21.23|50.53/21.23|55.21/21.23|60.57/21.23|63.96/21.24|
>
> SHE-LoRA reports two PPL metrics: the value on the left of `/` reflects the model performance when encrypted parameters are masked (i.e., using only unencrypted columns), while the value on the right of `/` reflects the model’s true performance without masking parameters.
> As expected, the `Base` model exhibits high PPL due to lack of domain adaptation, whereas `Vanilla LoRA` significantly reduces PPL, confirming effective learning.
> Notably, SHE-LoRA with an encryption ratio of merely 1‰ already raises the masked PPL to 38.83, indicating that even trivial removal of the most sensitive columns substantially degrades utility.
> In contrast, the PPL result on the right of `/` is nearly identical to that of `Vanilla LoRA`, demonstrating sound utility preservation under SHE-LoRA.
> Furthermore, as the encryption ratio increases, masked PPL consistently rises, confirming that SHE-LoRA prioritizes the most privacy-sensitive columns.
>
> >### **Weakness 4:** The writing of the paper can be further improved.
> > - Some notations and definitions are missing in the paper. For example, the notation of "batch" uses the same notation as the matrix $B$ in LoRA.
> > - Too many details are omitted in the main text, which makes readers hard to follow.
>
> **Response 4:** We sincerely appreciate your suggestions for improving the paper’s clarity.
> We have improved the writing by fixing inline citations and grammatical issues.
> And we added a notation table in `Appendix G` to clearly define and unify all symbols (e.g., deduplicating the notation of $B$ by not using it for batch size), which will help avoid ambiguity and improve readability.
>
> Additionally, we have moved key technical details to the main text to ensure the paper is self-contained and easier to follow.
> For example, the formal privacy guarantee of SHE-LoRA has been added to `Section 3.5`, the data partitioning method has been moved to `Section 4.1`.
> The relationship between sensitive parameters and privacy risks has been emphasized in`Section 2.1`.

---

### Official Review · Reviewer_pYmU · 2025-10-30

**Soundness:** 3
**Presentation:** 3
**Contribution:** 2
**Rating:** 4
**Confidence:** 3

**Summary:**

The paper proposes SHE-LoRA, a framework that combines selective homomorphic encryption (SHE) with low-rank adaptation (LoRA) for cross-device federated fine-tuning of LLMs. The core ideas are: (1) negotiation across heterogeneous clients using order-preserving encryption to choose columns to encrypt based on channel-wise sensitivity; (2) column swapping to cluster encrypted vs. plaintext columns for efficient batching and to obfuscate positions; (3) column-aware adaptive aggregation that aggregates plaintext and ciphertext parts separately; and (4) reparameterization via SVD so each client recovers LoRA factors at its local rank. Experiments show accuracy comparable to non-private heterogeneous LoRA while reducing communication by up to 99.71% and encryption time by up to 99.87% vs. full-HE baselines, and strong robustness to DAGER gradient inversion.

**Strengths:**

1. Massive reductions in HE time and bandwidth vs. full HE and MaskCrypt, with stable per-client times due to column clustering and budget control.
2. Comparable to non-private Flex-LoRA across GLUE/MMLU and vision tasks; sometimes better on subsets.

**Weaknesses:**

1. OPE preserves order information; while only rankings are revealed, the paper does not quantify leakage from order disclosure or compare with order-revealing encryption alternatives.
2. For 30B/70B models, ciphertext size/time per parameter increases notably and requires larger key sizes. While still workable at small budgets, the practicality at higher budgets or longer runs is unclear.

**Questions:**

1. How does SHE-LoRA fare against membership/property inference or reconstruction attacks that use auxiliary priors? Any reasons to expect similar robustness?
2. How should a,b,c be chosen in practice? Could you provide an adaptive rule and an ablation on real datasets?
3. For highly dynamic Non-IID clients, how often should renegotiation occur? Can you report end-to-end time or energy overheads for different periods?

---

> ### Author Response · Authors · 2025-11-23
> **Response (1/3)**
>
> We thank the reviewer for the insightful comments.
>
> ### Weaknesses
> >### **Weakness 1:** OPE preserves order information; while only rankings are revealed, the paper does not quantify leakage from order disclosure or compare with order-revealing encryption alternatives.
>
> **Weakness Response 1:** OPE is employed solely during the initial negotiation phase to identify which parameter columns are sensitive, and is never applied to actual parameter updates, which remain protected at all times by semantically secure HE.
> The disclosed column ordering reveals only the relative sensitivity ranking and does not expose raw parameter values, thereby preserving the confidentiality guarantees of HE.
> Moreover, SHE-LoRA is modular: if stronger order privacy is needed, OPE can be replaced by alternatives such as order-revealing encryption, secure multi-party computation, or a trusted third party.
> We have added the clarification of this characteristic in `Section 3.1.2` of the revised manuscript.
>
> >### **Weakness 2:** For 30B/70B models, ciphertext size/time per parameter increases notably and requires larger key sizes. While still workable at small budgets, the practicality at higher budgets or longer runs is unclear.
>
> **Weakness Response 2:** Indeed, ciphertext size and computational overhead grow with model scale (e.g., 30B/70B).
> However, SHE-LoRA targets resource-constrained devices characterized by heterogeneous data distributions and limited computation capabilities.
> These align with the scenarios targeted by SOTA non-privacy-preserving FL frameworks (e.g., HeterLoRA, Flex-LoRA), where most clients (e.g., edge devices) cannot afford full-model encryption or full fine-tuning due to hardware constraints.
>
> By enabling column-wise selective HE under strict encryption budgets, SHE-LoRA makes privacy-preserving federated LoRA feasible for resource-constrained clients.
> The same framework naturally scales to larger-scale deployments.
> For example, in enterprise settings with more capable hardware, increased encryption budgets allow more parameter columns to be protected, and hence enhancing privacy at the cost of additional computation and communication overhead.
> While HE costs do increase with model size, we believe such trade-offs will become increasingly acceptable as on-device capabilities improve and demand for private LLM adaptation rises.

---

> ### Author Response · Authors · 2025-11-23
> **Response (2/3)**
>
> ### Questions
>
> >### **Question 1:** How does SHE-LoRA fare against membership/property inference or reconstruction attacks that use auxiliary priors? Any reasons to expect similar robustness?
>
> **Response 1:** We fine-tuned the base model `Qwen3-4B-Instruct-2507` on the PILE dataset using standard LoRA (denoted as `Vanilla LoRA` in tables) and SHE-LoRA with varying encryption ratios $\gamma$, respectively.
> Then, we implemented seven membership inference attacks (MIA) on the base model (denoted as `Base` in tables) and the fine-tuned models of Vanilla LoRA and SHE-LoRA, and collect the attack results.
> Under SHE-LoRA, attackers can only launch MIA based on unencrypted parameters.
> Results are reported in AUROC, FPR@95, and TPR@5.
>
> Below are the AUROC results:
>
> |Model|Loss[c1]|Lowercase[c1]|Zlib[c1]|Min-k(0.1)[c2]|Min-k(0.5)[c2]|Recall[c3]|PAC[c4]|
> |-|-|-|-|-|-|-|-|
> |Base|50.9%|48.4%|50.2%|50.5%|50.9%|50.1%|51.2%|
> |Vanilla LoRA|81.4%|80.5%|76.7%|80.9%|82.9%|73.8%|83.3%|
> |$\gamma=1‰$|62.6%|62.8%|60.3%|62.5%|63.5%|64.7%|65.0%|
> |$\gamma=$ 1%|56.8%|57.7%|55.4%|56.5%|57.3%|58.4%|58.4%|
> |$\gamma=$ 5%|54.1%|55.2%|53.1%|53.7%|54.3%|55.8%|55.3%|
> |$\gamma=$ 10%|56.8%|57.7%|55.4%|56.5%|57.3%|58.4%|58.4%|
> |$\gamma=$ 20%|52.4%|53.2%|51.7%|52.1%|52.5%|53.1%|53.3%|
>
> In the AUROC results, `Base` performs no better than random guessing (with AUROC$\approx$50%), confirming that the pretraining corpus does not include the evaluation data.
> In contrast, `Vanilla LoRA` achieves much higher AUROC results across all attacks, indicating substantial membership leakage after fine-tuning.
> Remarkably, compared with `Vanilla LoRA`, SHE-LoRA reduces the average MIA success rate by 21.0% with an encryption ratio as low as $\gamma = 1‰$.
> With the increasing of $\gamma$ (e.g., to 1%), attack success rates further drop by 20.9%~30.9%, resulting in nearly random-guessing performance and demonstrating significantly stronger privacy protection.
>
> Below are the FPR@95 results:
>
> |Model|Loss|Lowercase|Zlib|Min-k(0.1)|Min-k(0.5)|Recall|PAC|
> |-|-|-|-|-|-|-|-|
> |Base|94.1%|95.4%|96.1%|96.1%|95.2%|95.5%|95.7%|
> |Vanilla LoRA|66.5%|63.3%|88.8%|71.4%|66.2%|79.2%|72.1%|
> |$\gamma=1‰$|89.7%|86.9%|93.9%|90.8%|90.3%|90.5%|89.9%|
> |$\gamma=$ 1%|92.4%|90.7%|95.2%|93.8%|93.4%|91.6%|93.1%|
> |$\gamma=$ 5%|93.2%|92.1%|95.5%|94.5%|94.1%|93.1%|93.8%|
> |$\gamma=$ 10%|92.4%|90.7%|95.2%|93.8%|93.4%|91.6%|93.1%|
> |$\gamma=$ 20%|93.2%|93.5%|95.6%|95.4%|95.1%|94.0%|94.8%|
>
> Below are the TPR@5 results:
>
> |Model|Loss|Lowercase|Zlib|Min-k(0.1)|Min-k(0.5)|Recall|PAC|
> |-|-|-|-|-|-|-|-|
> |Base|8.6%|5.6%|7.4%|5.5%|8.0%|5.1%|9.9%|
> |Vanilla LoRA|35.7%|35.6%|39.7%|39.4%|41.8%|25.4%|53.5%|
> |$\gamma=1‰$|12.5%|13.0%|15.2%|14.0%|14.5%|15.4%|17.4%|
> |$\gamma=$ 1%|9.7%|9.3%|10.8%|9.1%|10.2%|10.3%|15.0%|
> |$\gamma=$ 5%|9.0%|7.4%|9.1%|7.3%|9.3%|7.7%|12.4%|
> |$\gamma=$ 10%|9.7%|9.3%|10.8%|9.1%|10.2%|10.3%|11.9%|
> |$\gamma=$ 20%|8.2%|6.7%|8.5%|6.5%|8.2%|6.2%|11.1%|
>
> Consistent with the AUROC results, even at $\gamma = 1‰$, SHE-LoRA achieves an average FPR@95 of 90.27% and an average TPR@5 of 14.57%, closely comparable to the base model's performance (FPR@95=95.44%, TPR@5=7.16%).
> In contrast, `Vanilla LoRA` is significantly more vulnerable, with an average FPR@95 of 72.50% and TPR@5 of 38.73%.
> These results demonstrate that SHE-LoRA preserves membership privacy during fine-tuning: even under a very small encryption ratio, attackers can only achieve performance close to random guessing, with negligible advantage in distinguishing members from non-members.
>
> In summary, **experiments on `Qwen3-4B-Instruct-2507` across seven membership inference attacks demonstrate that SHE-LoRA is consistently robust**.
> This stems from two key design features: (1) selective encryption of the most sensitive parameter columns prevents direct leakage of privacy-critical information, and (2) column-wise position obfuscation, similar to injecting structured perturbations (see **Response 1 to Reviewer 9N3M**), increases uncertainty for attackers.
> These mechanisms also mitigate property inference and reconstruction attacks that leverage auxiliary priors, as they obscure the very gradients or parameters these attacks typically exploit.
>
> [c1] [Extracting Training Data from Large Language Models](https://www.usenix.org/system/files/sec21-carlini-extracting.pdf), USENIX Security Symposium 2021.
>
> [c2] [Detecting Pretraining Data from Large Language Models](https://openreview.net/forum?id=zWqr3MQuNs), ICLR 2024.
>
> [c3] [ReCaLL: Membership Inference via Relative Conditional Log-Likelihoods](https://aclanthology.org/2024.emnlp-main.493/), EMNLP 2024.
>
> [c4] [Data Contamination Calibration for Black-box LLMs](https://aclanthology.org/2024.findings-acl.644/), ACL 2024.

---

> ### Author Response · Authors · 2025-11-23
> **Response (3/3)**
>
> >### **Question 2:** How should a,b,c be chosen in practice? Could you provide an adaptive rule and an ablation on real datasets?
>
> **Response 2:** In practice, the coefficients $a$, $b$ and $c$ are determined via Bayesian optimization under the simplex constraint $a + b + c = 1$.
>
> Specifically, the negotiation objective is to reach a principled trade-off between **minimal client coverage** and **maximal privacy leakage** in the worst case, while keeping the total encryption budget fixed.
> - *Client coverage* is defined as the proportion of sensitive parameters per client that are included in the negotiated set.
> - *Privacy leakage* is quantified as the proportion of sensitivity exposed via unselected parameter columns per client.
>
> Accordingly, the negotiation goal is formulated as a bi-objective optimization problem that:
> - maximize the minimal client coverage of sensitive parameter columns, and
> - minimize the maximal privacy leakage of unselected parameter columns
>
> over all clients in the worst case.
>
> Given the constraint $c = 1 - a - b$, the optimization reduces to searching over the feasible domain $(a, b) \in [0,1]^2$ with $a + b \leq 1$. A Gaussian process-based Bayesian optimizer efficiently explores this domain, evaluates the composite objective score at each candidate $(a, b)$, and iteratively locates the optimal triple $(a^{\*}, b^{\*}, c^{\*})$.
>
> Further implementation details are provided in `Appendix D.1`.
>
> >### **Question 3:** For highly dynamic Non-IID clients, how often should renegotiation occur? Can you report end-to-end time or energy overheads for different periods?
>
> **Response 3:** As shown in Figure 11, the parameter-wise sensitivity distribution does change slightly, while the distribution of column-wise parameter sensitivity values remains relatively stable during FL training, as compared to Figure 4.
> Given that the parameter importance in SHE-LoRA is assessed via column-wise summation of sensitivity values (Lines 219-221), the slight change of parameter-wise sensitivity distribution generally has minor impact on performance.
> Therefore, SHE-LoRA does not trigger renegotiation automatically, but let clients decide the renegotiation plan according to their tolerance to sensitivity distribution change.
>
> Once triggered, the theoretical costs of negotiation and training per layer on a client are listed as below, which have also been included in `Appendix D.6`.
>
> ||Communication|Computation|
> |-|-|-|
> |Negotiation$\times$N|4 Bytes$\times$hidden size$\times$ratio$\times$N|Forward$\times$N|
> |Training$\times$1|2 Bytes$\times$rank$\times$hidden size|Backpropagation$\times$1|
>
> With bf16 precision, encryption ratio=1%, rank=16 and hidden size for `Llama-30B`=6656, FL training generally takes <50 rounds for convergence.
> Even if the negotiation is executed per round, the N-round negotiation communication overhead=4 Bytes$\times$6656$\times$1%$\times$50=13 KB << the training communication overhead=2 Bytes$\times$16$\times$6656=213 KB.
> However, although the computation cost of a `forward` << the cost of a `backpropagation`, *Forward$\times$N* will gradually increase along with the training progress.
> Therefore, **the clients can choose the negotiation period according to their expected balance between sensitivity update timeliness and computation cost**.

---

### Official Review · Reviewer_i41f · 2025-11-01

**Soundness:** 3
**Presentation:** 3
**Contribution:** 2
**Rating:** 6
**Confidence:** 2

**Summary:**

The proposed method SHE-LoRA integrates selective homomorphic encryption (SHE) with LoRA-based federated fine-tuning under heterogeneous client resources and Non-IID data. It aims to tackle the challenge of adaptive SHE for heterogeneous clients and the expansion of encrypted subsets on SHE. Specifically, this is done through the proposed HE subset negotiation mechanism and selective encryption and column-aware aggregation. Empirical results have shown that SHE-LoRA is resistant to privacy attacks, communication-efficient, and performant.

**Strengths:**

- The proposed method is empirically effective, which shows advantages in privacy-preserving, communication cost, and model performance.
- The problem of vulnerable and heterogeneous LoRA updates is motivated well.
- Principled design of the proposed algorithm.

**Weaknesses:**

- Limited novelty in the adoption of SHE methods to LLM LoRA fine-tuning.
- Column-wise weighted averaging is proposed, but the choice of weights (e.g., proportional to client data size or sensitivity) is not formally justified or compared.
- The negotiated global HE subset is claimed to optimally balance privacy and HE overhead per client, but lacks formal optimality guarantees or approximation bounds (e.g., submodular coverage, budgeted max coverage).
- The paper argues that encrypting A is sufficient, but the threat model and Section 2.4 discuss expanded encrypted subsets due to BA multiplication. It is not clear whether encrypting only A’s sensitive columns is enough under all considered attacks.
- Presentation could be improved by fixing the inline citations, defining the exact weighting scheme in column averaging, and reporting the Non-IID partitioning method.

**Questions:**

See weaknesses above.

---

> ### Author Response · Authors · 2025-11-23
> **Response (1/3)**
>
> Thank you for your valuable feedback.
>
> >### **Question 1:** Limited novelty in the adoption of SHE methods to LLM LoRA fine-tuning.
>
> **Response 1:** As stated in `Section 2.4` of the paper, naive adaptation of existing SHE methods to federated LoRA fine-tuning leads to ciphertext expansion (Figure 1) due to matrix multiplication, which significantly increases both cryptographic and communication overheads, rendering it impractical for resource-constrained devices.
> Moreover, heterogeneous clients may select different HE subsets for encryption, further exacerbating the inflation of ciphertext size (Figure 2).
> **These challenges remain unresolved prior to the introduction of SHE-LoRA.**
>
> To the best of our knowledge, SHE has not been previously applied to LoRA fine-tuning of LLMs in heterogeneous FL.
> Specifically, we firstly constrain ciphertext size expansion across heterogeneous clients, through `HE subset negotiation` (Section 3.1).
> Second, by leveraging a `column-aware encryption and swapping` mechanism (Section 3.2), we reduce computational overhead on ciphertexts and prevent ciphertext expansion induced by matrix multiplication.
> Third, `adaptive aggregation` (Section 3.3) separately aggregates plaintext and ciphertext model parameters, thereby preserving model utility while avoiding ciphertext inflation caused by matrix addition from heterogeneous clients.
> Finally, a customized `reparameterization` technique (Section 3.4) enables the reconstruction of client-specific heterogeneous LoRA parameters from the aggregated result.
>
> Through these innovations, **SHE-LoRA enables the practical integration of SHE into federated LoRA with heterogeneous device capabilities, while preserving model performance and providing strong privacy protection (recognized by Reviewers `9N3M`, `pYmU` and `sB8q`).**
> Crucially, it mitigates the otherwise prohibitive cryptographic and communication overheads, making privacy-preserving federated PEFT feasible even for resource-constrained clients.
> The effectiveness of these innovations have been validated across multiple models, tasks, and attack scenarios (reconstruction and membership inference attacks) in our experiments.
>
> >### **Question 2:** Column-wise weighted averaging is proposed, but the choice of weights (e.g., proportional to client data size or sensitivity) is not formally justified or compared.
>
> **Response 2:** SHE-LoRA aims to integrate SHE into general federated LLM PEFT for privacy preservation guarantee, the determination of weight per client during model aggregation can be adjusted according to the design of FL frameworks (e.g., weight-averaged by client data size as in FedAvg [c1]).
> When compared with common SOTA non-privacy-preserving FL frameworks (e.g., HeterLoRA, Flex-LoRA), the main uniqueness of SHE-LoRA lies in the separate aggregation of ciphertext and plaintext.
> The rationale to this design is three-fold:
> 1. To **avoid the inflation of ciphertext size** caused by adding encrypted and unencrypted model parameters (Figure 2).
> 2. Since contributing clients vary per column, column-wise weighted averaging can seamlessly adopt well-established aggregation schemes such as FedAvg.
> 3. The column-wise **aggregated results** of ciphertext and plaintext **can be finally combined without altering the fine-tuning paradigm of LoRA or degrading model utility** as proved in `Appendix D.4` and `Appendix D.5`.
>
> In summary, the column-wise weighted averaging in SHE-LoRA is designed to avoid the inflation of ciphertext size while adapting to the weight-averaging strategy of general federated LLM PEFT frameworks.
>
> [c1]. [Communication-Efficient Learning of Deep Networks from Decentralized Data](https://arxiv.org/abs/1602.05629), arXiv:1602.05629.

---

> ### Author Response · Authors · 2025-11-23
> **Response (2/3)**
>
> >### **Question 3:** The negotiated global HE subset is claimed to optimally balance privacy and HE overhead per client, but lacks formal optimality guarantees or approximation bounds (e.g., submodular coverage, budgeted max coverage).
>
> **Response 3:** To ensure the optimality of HE subset negotiation, resource-constrained clients must inevitably trade off between **minimal client coverage of sensitive parameters** and **maximal privacy leakage of unencrypted parameters** under limited HE budgets.
> The negotiation objective is to reach a balance that extends HE-based privacy protection to as many clients as possible, while minimizing the privacy leakage from each client’s unselected parameters.
> We formulate this as a bi-objective optimization problem and provide its detailed solution in `Appendix D.1`.
>
> Moreover, as detailed in **Response 1 to Reviewer 9N3M**, we also provide a `theoretical lower bound` on the reconstruction error achievable by an adversary attempting to infer private training data from the unencrypted parameters:
>
> $$
> {E}_\mathcal{R}\geq\frac{d^2}{\mathbb{E}\_{x\sim\mathcal{X}}[\text{tr}(J\_F(x))]+\lambda_e(J\_P)}\geq\frac{d^2}{\frac{n(1-\gamma)}{s^2}\mathbb{E}\_{x \sim \mathcal{X}}||\nabla\_xg(x)||\_{\max}^2+\lambda_e(J\_P)} \tag{1}
> $$
> where $J\_F(x)$ is the Fisher information matrix, $\text{tr}(\cdot)$ is the trace operator, $\lambda\_e(J_P)$ is the largest eigenvalue of the prior-informed Fisher information matrix $J_P$, $d$ is the data dimension, $n$ is the number of columns in $\mathbf{G}$, $\gamma$ is the encryption ratio, $s^2$ is the variance of the equivalent noise, and $||\nabla\_xg(x) ||\_{\max}^2 = (\max\_{i,j}\|\nabla\_x g\_j(x\_i)\|)^2$ quantifies the maximum gradient exposure, which is defined as the squared maximum sensitivity of an unencrypted (exposed) gradient coordinate $g_j(x_i)$ with respect to a data feature $x_i$.
>
> As Equation (1) shows, the reconstruction error $E\_\mathcal{R}$ is lower-bounded by a quantity whose denominator depends on $\text{tr}(J\_F(x))$ and $\lambda\_e(J_P)$.
> Since $\lambda\_e(J_P)$ is determined solely by the data prior, the bound under fixed $n$, $s^2$, and $\gamma$ is governed exclusively by $||\nabla\_xg(x)||\_{\max}^2$.
> By selectively encrypting the most sensitive columns, which are measured in terms of the Wanda parameter sensitivity (a proxy for their contribution to input-space sensitivity via the score $S_j$ in Line 217), SHE-LoRA suppresses the dominant terms in the Fisher information matirx, thereby reducing the magnitude of the unencrypted gradients $\|\nabla_xg(x) \|_{\max}^2$ and lowering $\text{tr}(J_F(x))$.
> Thus, **by encrypting the most sensitive parameters, SHE-LoRA increases the minimum achievable reconstruction error and strengthens privacy against any gradient inversion attack**.

---

> ### Author Response · Authors · 2025-11-23
> **Response (3/3)**
>
> >### **Question 4:** The paper argues that encrypting A is sufficient, but the threat model and Section 2.4 discuss expanded encrypted subsets due to BA multiplication. It is not clear whether encrypting only A’s sensitive columns is enough under all considered attacks.
>
> **Response 4:** First, the ciphertext expansion induced by matrix multiplication $BA$ (Section 2.4), a direct consequence of naively applying SHE to LoRA, motivates the design of SHE-LoRA.
> This issue not only inflates cryptographic and communication costs, but also undermines practicality in resource-constrained federated settings, highlighting the need for a more cost-effective and secure way to integrate HE into Federated LLM PEFT.
>
> Second, the forward pass is given by $Y^{L\times m} =X W\_0^\top +X (B A)^\top$,
> where $X \in \mathbb{R}^{L \times n}$ denotes the input embeddings with $L$ tokens of dimension $n$, $A \in \mathbb{R}^{r \times n}$, and $B \in \mathbb{R}^{m \times r}$ with $r \ll \min(m,n)$.
> When the pre-trained weight $W\_0\in \mathbb{R}^{m\times n}$ is frozen, the gradients of the loss $\mathcal{L}$ with respect to $A$ and $B$ are:
>
> $$
> (\frac{\partial \mathcal{L}}{\partial A})^{r\times n} = (B^\top)^{r\times m}( \frac{\partial \mathcal{L}}{\partial Y}^\top)^{m\times L} X^{L\times n}, \quad
> (\frac{\partial \mathcal{L}}{\partial B})^{m\times r} = (\frac{\partial \mathcal{L}}{\partial Y}^\top)^{m\times L} X^{L\times n} (A^\top)^{n\times r}.
> $$
>
> Crucially, the gradient of $A$ contains the full input $X$, modulated only by the fixed low-dimensional transform $B^\top$. As $B^\top$ does not compress $X$, this gradient preserves nearly all information of the input. In contrast, the gradient of $B$ can receive $X$ only after its projection through the narrow bottleneck $A^\top$. Given the low rank $r \ll n$, this projection drastically limits the amount of recoverable information in $X$.
>
> Consequently, an adversary can reconstruct $X$ far more accurately from $\frac{\partial \mathcal{L}}{\partial A}$ than from $\frac{\partial \mathcal{L}}{\partial B}$ [c1].
> This asymmetry justifies our design choice: under a limited HE budget, we prioritize encrypting the sensitive columns from $A$ matrix.
>
> [c1] [DAGER: Exact Gradient Inversion for Large Language Models](https://openreview.net/forum?id=CrADAX7h23&referrer=[the%20profile%20of%20Martin%20Vechev]%28%2Fprofile%3Fid%3D~Martin_Vechev1%29), NeurIPS 2024
>
> >### **Question 5:** Presentation could be improved by fixing the inline citations, defining the exact weighting scheme in column averaging, and reporting the Non-IID partitioning method.
>
> **Response 5:**
> We have improved the writing by fixing inline citations and grammatical issues.
> Additionally, as noted in **Response 2**, the weighting scheme used in column-wise averaging can be adjusted according to the design of FL frameworks, and we have clarified this in `Section 3.3`.
> Finally, we acknowledge that the description of heterogeneous data partitioning (e.g., Dirichlet with $\rho = 0.3$) was inadvertently placed in the appendix, and we have reported this key technical detail in `Section 4.1`.

---

### Official Review · Reviewer_9N3M · 2025-11-04

**Soundness:** 3
**Presentation:** 2
**Contribution:** 3
**Rating:** 6
**Confidence:** 4

**Summary:**

The paper proposes a privacy-preserving method for fine-tuning large language models in federated learning. Traditional LoRA-based methods risk data leakage through shared gradients, while full homomorphic encryption is too costly. SHE-LoRA introduces selective homomorphic encryption and designs a column-level negotiation and aggregation mechanism to handle heterogeneous clients with different computing resources and privacy needs. The approach includes four key stages: (1) clients privately report column sensitivity to jointly select a global subset for encryption; (2) encrypted columns are reordered for efficient HE operations; (3) aggregation aligns plaintext and ciphertext parts column-wise; (4) results are re-factorized into new low-rank parameters. Experiments on NLP tasks show SHE-LoRA maintains model accuracy while reducing encryption and communication overhead by up to 99% compared with full HE, and it effectively defends against gradient inversion attacks like DAGER. The system scales to large models and non-IID data, proving that partial, structured encryption can offer strong privacy with minimal cost in federated LoRA training.

**Strengths:**

1. The framework explicitly supports clients with different hardware capabilities, network conditions, and privacy budgets.
2. This paper introduces selective homomorphic encryption at the column level of LoRA matrices, encrypting only the most privacy-sensitive components.
3. Experiments on NLP and vision datasets demonstrate that SHE-LoRA achieves accuracy comparable to or better than state-of-the-art methods (e.g., Flex-LoRA) under heterogeneous and Non-IID conditions.

**Weaknesses:**

1. The privacy guarantees of selective encryption, the convergence behavior of federated training under mixed plaintext/ciphertext updates, and the optimality of the HE subset negotiation are not formally proved.
2. SHE-LoRA method can't adapt to heterogeneous LoRA approaches like FLoRA.
3. The experiments rely on relatively small base models and simple benchmark tasks, which limits the generalizability of its results to large-scale or more complex real-world scenarios. It is recommended to evaluate the method on stronger base models such as Qwen3 and Llama 3.2, as well as on more challenging benchmark tasks including MMLU-Pro, GPQA, MuSR, MATH, IFEval, and BBH.

**Questions:**

See Weakness.

---

> ### Author Response · Authors · 2025-11-23
> **Response (1/2)**
>
> Thank you for the valuable feedback.
>
> >### **Question 1:** The privacy guarantees of selective encryption, the convergence behavior of federated training under mixed plaintext/ciphertext updates, and the optimality of the HE subset negotiation are not formally proved.
>
> **Response 1:**
>
> **Privacy guarantee of selective encryption**
>
> ---
>
> First, the parameters protected by HE do not leak privacy, and privacy risks are mainly caused by unencrypted parameters.
> The reconstruction error $E_\mathcal{R}$ can be defined as the minimum expected squared reconstruction error [c1]:
> $$
> {E}\_\mathcal{R}=\min_{\mathcal{R}} \mathbb{E}\_{x\sim\mathcal{X}} \mathbb{E}\_{\mathbf{y}\sim f(g(x))}\Bigl[\bigl||\mathcal{R}(\mathbf{y}) - x\bigr||\_2^2\Bigr] \tag{1}
> $$
> where $\mathcal{R}(\cdot)$ is any data reconstruction attack method, $f(\cdot)$ is the selective HE method of SHE-LoRA, and $g(x)$ is the gradient calculated on data $x$.
>
> Second, the column swapping process, which serves as column-wise obfuscation, increases the difficulty of privacy attacks, and can be viewed as adding an asymptotic Gaussian-distributed noise (with $s^2$ as the equivalent variance) to the original gradient as detailed in `Appendix D.7`.
>
> Thus, according to [c2], for any reconstruction algorithms that aim to infer private data from observing noisy and partially encrypted model updates, the lower bound of $E_\mathcal{R}$ can be given by the `Bayesian Cramér-Rao Lower Bound` as:
> $$
> {E}\_\mathcal{R}\geq\frac{d^2}{\mathbb{E}\_{x\sim\mathcal{X}}[\text{tr}(J\_F(x))]+ \lambda\_e(J\_P)}\geq\frac{d^2}{\frac{n(1-\gamma)}{s^2}\mathbb{E}\_{x \sim \mathcal{X}} ||\nabla\_xg(x) ||\_{\max}^2+ \lambda_e(J\_P)} \tag{2}
> $$
> where $J\_F(x)$ is the Fisher information matrix, $\text{tr}(\cdot)$ is the trace operator, $\lambda\_e(J_P)$ is the largest eigenvalue of the prior-informed Fisher information matrix $J_P$, $d$ is the data dimension, $n$ is the number of columns in $\mathbf{G}$, $\gamma$ is the encryption ratio, $s^2$ is the variance of the equivalent noise, and $||\nabla\_xg(x) ||\_{\max}^2 = (\max\_{i,j}\|\nabla\_x g\_j(x\_i)\|)^2$ quantifies the maximum gradient exposure, which is defined as the squared maximum sensitivity of an unencrypted (exposed) gradient coordinate $g_j(x_i)$ with respect to a data feature $x_i$.
>
> Equation (2) means that the reconstruction error $E_\mathcal{R}$ is lower-bounded by a quantity whose denominator depends on $\text{tr}(J\_F(x))$ and $\lambda\_e(J\_P)$.
> Since $\lambda\_e(J\_P)$ is determined solely by the data prior, the bound under fixed $n$, $s^2$, and $\gamma$ is governed exclusively by $||\nabla\_xg(x) ||\_{\max}^2$.
> By selectively encrypting the most sensitive columns, which are measured in terms of the Wanda parameter sensitivity (a proxy for their contribution to input-space sensitivity via the score $S_j$ in Line 217), SHE-LoRA suppresses the dominant terms in the Fisher information matrix, thereby reducing the magnitude of the unencrypted gradients $\|\nabla_xg(x) \|_{\max}^2$ and lowering $\text{tr}(J_F(x))$.
>
> In summary, **by encrypting the most sensitive parameters, SHE-LoRA increases the minimum achievable reconstruction error and strengthens privacy against any gradient inversion attack**.
>
> [c1] [Optimal Defenses Against Data Reconstruction Attacks](https://openreview.net/forum?id=5idgXsjpPb), ICML 2025 Workshop DIG-BUG.
>
> [c2] [Advancing Practical Homomorphic Encryption for Federated Learning: Theoretical Guarantees and Efficiency Optimizations
> ](https://arxiv.org/abs/2509.20476), arXiv:2509.20476.
>
> **Convergence behavior of federated training under mixed plaintext/ciphertext updates**
>
> ---
> Second, as proved in `Appendix D.4` and `Appendix D.5`, the mixed plaintext/ciphertext update of model parameters does not alter the fine-tuning paradigm of LoRA, and hence preserves model utility and convergence behavior.
> Moreover, as shown in Tables 7, 8, 9 of the manuscript and **Response 3** below, SHE-LoRA achieves model performance comparable to the plaintext baseline under the same training rounds, which further validates the model utility losslessness characteristic of SHE-LoRA.
>
> **Optimality of the HE subset negotiation**
>
> ---
> Third, to ensure the optimality of HE subset negotiation, resource-constrained clients must inevitably trade off between **minimal client coverage of sensitive parameters** and **maximal privacy leakage of unencrypted parameters** under limited HE budgets.
> The negotiation objective is to reach a balance that extends HE-based privacy protection to as many clients as possible, while minimizing the privacy leakage from each client’s unselected parameters.
> We formulate this as a bi-objective optimization problem and provide its detailed solution in `Appendix D.1`.

---

> ### Author Response · Authors · 2025-11-23
> **Response (2/2)**
>
> >### **Question 2:** SHE-LoRA method can't adapt to heterogeneous LoRA approaches like FLoRA.
>
> **Response 2:** We emphasize that SHE-LoRA is designed for federated PEFT with heterogeneous LoRA.
> It allows each client to set different LoRA ranks and HE budgets as shown in Figure 3.
> The `HE subset negotiation` (Sec. 3.1) is specifically designed to enable heterogeneous clients to agree upon a global encrypted subset.
> The server eliminates the impact of heterogeneous LoRA ranks by computing $\Delta W = BA$ and aggregating the resulting $\Delta W$ in the process of `adaptive aggregation` (Sec. 3.3).
> Finally, a customized `reparameterization` (Sec. 3.4) technique is employed to reparameterize the aggregated result back into heterogeneous LoRA parameters.
> As a result, SHE-LoRA can adapt to heterogeneous LoRA approaches.
>
> For example, SHE-LoRA can easily adapt to FLoRA, which performs aggregation as $\sum\_{k=0}^{K} B\_kA\_k=(B\_{0}\oplus\dots\oplus B\_{K})(A\_{0}\oplus\dots\oplus A\_{K})$, by trivial modification of aggregation weights:
> ```
> // FLoRA
> delta_W += Bi * Ai * weights_dataset
> // SHE-LoRA
> delta_W += Bi * Ai * weights_dataset * weights_columns
> ```
> where `weights_dataset` is the weight normalized based on the amount of data on the client, and `weights_columns` is dependent on the number of clients that can contribute to the aggregation of the columns (e.g. Figure 6 in manuscript).
> >### **Question 3:** The experiments rely on relatively small base models and simple benchmark tasks, which limits the generalizability of its results to large-scale or more complex real-world scenarios. It is recommended to evaluate the method on stronger base models such as Qwen3 and Llama 3.2, as well as on more challenging benchmark tasks including MMLU-Pro, GPQA, MuSR, MATH, IFEval, and BBH.
>
> **Response 3:** Per the reviewer’s suggestion, we added the results of SHE-LoRA, Vanilla LoRA, and the base model, which are collected from the training of `Qwen3-4B-Instruct-2507` and `Llama-3.2-3B` on the PILE dataset, and evaluation on more challenging benchmarks.
>
> |Qwen3-4B|MMLU-Pro|GPQA|MuSR|MATH|IFEval|BBH|
> |-|-|-|-|-|-|-|
> |SHE-LoRA|47.36|43.94|30.71|78.86|67.34|63.26|
> |Vanilla LoRA|48.54|42.57|32.31|77.63|68.06|64.58|
> |Base|65.36|45.00|61.67|84.00|90.17|85.93|
>
> |Llama-3.2-3B|MMLU-Pro|GPQA|MuSR|MATH|IFEval|BBH|
> |-|-|-|-|-|-|-|
> |SHE-LoRA|14.86|13.64|17.86|21.74|27.24|9.26|
> |Vanilla LoRA|14.62|13.87|18.05|20.96|27.37|9.38|
> |Base|14.29|11.11|30.95|16.68|27.99|10.85|
>
> These results demonstrate that SHE-LoRA achieves the performance comparable to Vanilla LoRA, while providing privacy via selective HE.
> This benefit generalizes across stronger base model families and diverse tasks, provided that federated LoRA is used for PEFT.
> More results on SHE-LoRA’s scalability to larger models are presented in `Appendix E.2`.

---

### Author Response · Authors · 2025-11-23
**General Reply**

We sincerely thank all the reviewers for their effort, feedback, and comments.
We are pleased that the paper has been recognized for its cost effectiveness in reducing HE overhead while maintaining model utility (`9N3M`, `i41f`, `pYmU`, `sB8q`), insight about selective HE of important model parameters (`9N3M`, `sB8q`), practicality in supporting heterogeneous clients (`9N3M`, `i41f`, `pYmU`), and strong privacy protection against gradient inversion attacks like DAGER (`9N3M`, `sB8q`).

We invite each reviewer to consult our individual responses, where we address all points raised in detail.
In this general reply, we provide a summary of the primary changes performed in light of the feedbacks:

- **Privacy guarantee** (`9N3M`, `i41f`, `sB8q`)

    We provided a theoretical lower bound on the reconstruction error achievable by an adversary attempting to infer private training data from the unencrypted parameters (**Section 3.5, Appendix D.7**).

- **Necessity of renegotiation** (`pYmU`, `sB8q`)

    Through observing and confirming the stability of parameter sensitivity distribution during fine-tuning, we elaborated why SHE-LoRA does not trigger renegotiation automatically, but let clients decide the renegotiation plan according to their tolerance to sensitivity distribution change (**Appendix D.6**).

- **Optimality of HE subset negotiation** (`9N3M`, `pYmU`, `sB8q`)

    We added an explanation to show that the optimality of HE subset negotiation embodies a worst-case trade-off, which is achieved by formulating a bi-objective optimization problem that aims to extend HE-based privacy protection to as many clients as possible, while minimizing the privacy leakage from each client’s unselected parameters (**Appendix D.1**).

- **Evaluation on resistance against membership inference attack** (`pYmU`, `sB8q`)

    We added the defense scores of SHE-LoRA against seven membership inference attacks (**Appendix F.4**).

- **Evaluation on stronger base models and more challenging benchmarks** (`9N3M`)

    We added the results of SHE-LoRA and comparison methods, which are collected from the training of `Qwen3-4B-Instruct-2507` and `Llama-3.2-3B`, and evaluation on more challenging benchmarks (**Appendix E.3**).

- **Evaluation on the impact of sensitive parameters on performance** (`sB8q`)

    We added an ablation experiment by directly removing the most sensitive parameters to evaluate their impact on model performance. (**Appendix F.5**)

Given the extensive additional experiments and the detailed theoretical clarifications for addressing the reviewers’ questions, we have substantially revised the manuscript and highlighted the changes in blue.

---

### Meta-Review · Area_Chair_sbHw · 2026-01-08

**Summary:**

The privacy guarantees of selective encryption, the convergence behavior of federated training under mixed plaintext/ciphertext updates, and the optimality of the HE subset negotiation are not formally proved. SHE-LoRA method can't adapt to heterogeneous LoRA approaches like FLoRA. The experiments rely on relatively small base models and simple benchmark tasks. Limited novelty in the adoption of SHE methods to LLM LoRA fine-tuning. The choice of weights (e.g., proportional to client data size or sensitivity) is not formally justified or compared. The paper argues that encrypting A is sufficient, but the threat model and Section 2.4 discuss expanded encrypted subsets due to BA multiplication. For highly dynamic Non-IID clients, how often should renegotiation occur? Can you report end-to-end time or energy overheads for different periods? Given that the encrypted parameters are only a very small fraction, it is unclear whether directly removing those parameters from training can also produce similar model performance.

**Reviewer Concerns:**

Most of the concerns are addressed fully in the rebuttal.

**Reviewer Scores:**

Given the quality of the rebuttal, I believe the reviewers would have raised their scores.

---

### Decision · Program_Chairs · 2026-01-26

Accept (Poster)